

# Proteomics and bioinformatics analysis reveal potential roles of cadmium-binding proteins in cadmium tolerance and accumulation of *Enterobacter cloacae*

Kitipong Chuanboon[1,2], Piyada Na Nakorn[3], Supitcha Pannengpetch[2], Vishuda Laengsri[2], Pornlada Nuchnoi[2], Chartchalerm Isarankura-Na-Ayudhya[3] and Patcharee Isarankura-Na-Ayudhya[1]

[1] Department of Medical Technology and Graduate Program in Biomedical Sciences, Faculty of Allied Health Sciences, Thammasat University, Pathumthani, Thailand
[2] Center for Research and Innovation, Faculty of Medical Technology, Mahidol University, Bangkok, Thailand
[3] Department of Clinical Microbiology and Applied Technology, Faculty of Medical Technology, Mahidol University, Bangkok, Thailand

Corresponding author
Patcharee Isarankura-Na-Ayudhya, pujonoy@hotmail.com

## ABSTRACT

**Background**. *Enterobacter cloacae* (EC) is a Gram-negative bacterium that has been utilized extensively in biotechnological and environmental science applications, possibly because of its high capability for adapting itself and surviving in hazardous conditions. A search for the EC from agricultural and industrial areas that possesses high capability to tolerate and/or accumulate cadmium ions has been conducted in this study. Plausible mechanisms of cellular adaptations in the presence of toxic cadmium have also been proposed.

**Methods**. Nine strains of EC were isolated and subsequently identified by biochemical characterization and MALDI-Biotyper. Minimum inhibitory concentrations (MICs) against cadmium, zinc and copper ions were determined by agar dilution method. Growth tolerance against cadmium ions was spectrophotometrically monitored at 600 nm. Cadmium accumulation at both cellular and protein levels was investigated using atomic absorption spectrophotometer. Proteomics analysis by 2D-DIGE in conjunction with protein identification by QTOF-LC-MS/MS was used to study differentially expressed proteins between the tolerant and intolerant strains as consequences of cadmium exposure. Expression of such proteins was confirmed by quantitative reverse transcription-polymerase chain reaction (qRT-PCR). Bioinformatics tools were applied to propose the functional roles of cadmium-binding protein and its association in cadmium tolerance mechanisms.

**Results**. The cadmium-tolerant strain (EC01) and intolerant strain (EC07) with the MICs of 1.6 and 0.4 mM, respectively, were isolated. The whole cell lysate of EC01 exhibited approximately two-fold higher in cadmium binding capability than those of the EC07 and ATCC 13047, possibly by the expression of Cd-binding proteins. Our proteomics analysis revealed the higher expression of DUF326-like domain (a high cysteine-rich protein) of up to 220 fold in the EC01 than that of the EC07. Confirmation of the transcription level of this gene by qRT-PCR revealed a 14-fold induction in the EC01. Regulation of the DUF326-like domain in EC01 was more pronounced to

mediate rapid cadmium accumulation (in 6 h) and tolerance than the other resistance mechanisms found in the ATCC 13047 and the EC07 strains. The only one major responsive protein against toxic cadmium found in these three strains belonged to an antioxidative enzyme, namely catalase. The unique proteins found in the ATCC 13047 and EC07 were identified as two groups: (i) ATP synthase subunit alpha, putative hydrolase and superoxide dismutase and (ii) OmpX, protein YciF, OmpC porin, DNA protection during starvation protein, and TrpR binding protein WrbA, respectively. **Conclusion**. All these findings gain insights not only into the molecular mechanisms of cadmium tolerance in EC but also open up a high feasibility to apply the newly discovered DUF326-like domain as cadmium biosorbents for environmental remediation in the future.

# INTRODUCTION

Environmental contamination by toxic heavy metals has increased dramatically due to the consequence of global industrialization. Cadmium is one kind of heavy metal that exerts its detrimental toxicity to human, animals, plants and microorganisms (*Jaiswal, Verma & Jaiswal, 2018*). Cellular adaptation of bacteria to survive under cadmium exposure can be accounted as follows: reduction of cadmium uptake, cadmium sequestration or complexation, cadmium efflux, and secretion of extracellular polysaccharides (*Nies, 1992*; *Nies, 1999*; *Xu et al., 2017*). Among the other environmental bacteria, *Enterobacter cloacae* (EC), a Gram-negative bacillus belonging to the family of *Enterobacteriaceae*, can localize in gut and in various environmental conditions. The EC has been reported to be tolerant against many kinds of heavy metals and antibiotics. For instance, the standard strain of EC (ATCC 13047) has been found to carry many heavy-metal resistance genes belonging to 7 operons (*sil*, *ars*, *mer* and *cop*) (*Ren et al., 2010*). Moreover, overexpression of extracellular biological components (e.g., exopolysaccharide) has been found to play major roles in chromate resistance by preventing the permeability of chromate into the cells (*Iyer, Mody & Jha, 2004*; *Yang et al., 2007*). Some strain of EC that conferred lead resistance has been discovered to have increased production of exopolysaccharide (*Naik, Pandey & Dubey, 2012*). Accumulation of nickel and vanadium as well as the association between vanadium and multidrug resistance have been reported (*Hernandez, Mellado & Martinez, 1998*). Co-existence of heavy metal tolerance (*sil* operon involved in acquired silver resistance) and antibiotic resistance (blaCTX-M and blaKPC acquired extended-spectrum cephalosporin and carbapenem resistance) genes has recently been documented in EC complex (*Andrade et al., 2018*). With respect to the high tolerance and/or accumulation of metal ions, the EC has been used in many biotechnological and environmental science applications such as biohydrogen production (*Khanna et al., 2011*), heavy metal bioremediation (*Rahman et al., 2015*; *Xu et al., 2017*), nanoparticles-based

biotransformation, biocatalyst for decolourization of azo dye (*Prasad & Aikat, 2014*). Redox transformation of toxic selenium oxyanions by the twin arginine translocation system has been discovered (*Ma, Kobayashi & Yee, 2007*). However, it is noteworthy that the resistance mechanisms against toxic cadmium ions in *Enterobacter cloacae* remain unclear.

Domains of unknown/uncharacterized function (DUFs) have been recognized as a series of uncharacterized protein families, which can be accessed through the international database namely Pfam database (available at https://pfam.xfam.org/) (*Bateman, Coggill & Finn, 2010*; *Finn et al., 2008*). Currently, the number of protein families has dramatically been reported of almost upto 18,000 entries (*El-Gebali et al., 2019*). Identification of the DUFs that occur within the protein gains a better understanding on the function of proteins found in nature. The DUFs seem to be essential not only for the normal cellular function (*Mashruwala & Boyd, 2018*) but they also become more important, particularly under certain conditions such as stress responses, biofilm formation, pathogenesis, intracellular metal homeostasis and environmental adaptation (*Eletsky et al., 2014*; *Guo et al., 2016*; *Mashruwala & Boyd, 2018*; *Tong et al., 2016*). Many of the DUFs are conserved and shared among prokaryotes and eukaryotes. For instances, the DUF59 has been documented to involve in intracellular Fe homeostasis in most of the organisms (*Mashruwala & Boyd, 2018*). The DUF2233, a protein domain with phosphodiester glycosidase activity, can be found in many bacteria, viruses and mammalian cells (*Das et al., 2013*). Even though some of the DUFs (e.g., DUF2525, DUF2526 and DUF2545) have been conserved in many bacteria belonging to the family of *Enterobacteriaceae*, little is known about the regulation of DUFs in *Enterobacter* spp. Moreover, the role of DUFs on metal resistance mechanisms of the EC has never been explored.

In this study, isolation of *Enterobacter cloacae* from agricultural and industrial sites has been performed in order to search for the environmental isolates that possessed ability to tolerate and/or accumulate cadmium ions. Minimum inhibitory concentrations (MICs) together with binding capability to cadmium ions have been analyzed. Alterations of protein expression profiles upon exposure to toxic cadmium have subsequently been investigated using two-dimensional-difference in gel electrophoresis (2D-DIGE) in conjunction with protein identification *via* liquid chromatography-mass spectrometry (LC-MS). Differentially-expressed proteins among different strains of EC as consequences of cadmium exposure have been analyzed and confirmed using qRT-PCR. In addition, molecular structure and functions together with their association networks have been determined by bioinformatics tools. Plausible explanations on the underlying mechanisms on how the EC respond and adapt themselves in survival against harmful cadmium have been discussed.

## MATERIALS & METHODS

### Isolation and identification of bacteria from water and soil around agricultural and industrial sites

Nine strains of bacteria were isolated from soil and water in various sites around agricultural and industrial areas in the central part of Thailand. Genus and species identification of these

isolates was performed by Gram's staining and standard biochemical testings including triple sugar iron agar (TSI), lysine iron agar (LIA), sulfide-indole-motility (SIM), citrate and urease. These bacterial colonies were also subjected to MALDI/TOF mass spectrometry for further confirmation as follows. Isolated colonies were analyzed using a formic acid–based direct on-plate preparation method. One microliter of 70% formic acid (Fluka, Sigma-Aldrich, St. Louis, MO, USA) per well was deposited onto the MALDI-TOF-MS steel anchor plate (BigAnchor 96-well plate; Bruker Daltonics, Fremont, CA, USA). Colonies were smeared into the formic acid and allowed to dry. The dried mixture was overlaid with 2 μl of matrix solution ($\alpha$-cyano-4-hydroxycinnamic acid; HCCA) dissolved in 50% acetonitrile, 47.5% water, and 2.5% trifluoroacetic acid (Fluka; Sigma-Aldrich) and allowed to dry prior to analysis using the MALDI Biotyper (Bruker Daltonics). A bacterial test standard (BTS; Bruker Daltonics) was used for instrument calibration. A positive control (*Staphylococcus aureus* ATCC 25923) and a negative control (formic acid and matrix) were included in each run. A MicroFlex LT mass spectrometer (Bruker Daltonics) was used for spectral analysis. Spectra were analyzed using the Bruker Biotyper 3.0 software and library version 3.3.1.0 (4,613 entries), supplemented with mass spectra from an in-house collection of 87 anaerobic isolates encompassing 39 species. Manufacturer-recommended cut-off scores were used for identification, with scores of $\geq 2.000$ indicating identification to the species level, scores between 1.700 and 1.999 indicating identification to the genus level, and scores of <1.700 indicating no identification. Isolates producing scores of <1.700 were retested once with the highest score used for final analysis. These nine isolates of *Enterobacter cloacae* were further deposited at $-80\,°C$ in the repository room (deposition reference numbers: EC_B-Cd-A series) at the Faculty of Medical Technology, Mahidol University, Salaya campus, Nakornpathom, Thailand.

## Determination of minimum inhibitory concentration (MIC) by agar dilution

Determination of MIC by agar dilution was performed according to a standard guideline from the Clinical and Laboratory Standards Institute, 2012 (CLSI, 2012). Briefly, *E. cloacae* was incubated in the Luria-Bertani (LB) broth (10 g/L tryptone, 5 g/L NaCl and 5 g/L yeast extract, pH 7.2) at 37 °C for overnight. Then, cells were collected and adjusted to be equivalent to 0.5 McFarland standard. Then, 2 μl of ten-fold diluted of 0.5 McFarland suspension was spotted onto each plate spreaded with final concentrations of cadmium, copper and zinc at 3.2–0.1 mM, 16-1 mM, and 6.4–0.1 mM, respectively. These plates were further incubated at 37 °C for 18 h prior to the MIC analysis.

## Growth curves determination in the presence of sub-lethal dose of cadmium ions

To observe the effect of cadmium ions on bacterial cells growth, overnight cultures of cells in 5 ml LB were adjusted to $OD_{600}$ of 0.001 in a new tube containing 3 ml LB broth. Cells were grown at 37 °C with shaking at 200 rpm for 30 min prior to addition of $CdCl_2$ to yield the final concentration of 0.2 mM. Each of the bacterial isolates was taken into three replicates. These tubes were incubated in shaking incubator at 37 °C at 200 rpm. Growth of cells was monitored by a spectrophotometer at the time interval of 1 h for 24 h. It is

noteworthy that the sub-lethal dose of 0.2 mM cadmium ions was equivalent to MIC/2 of intolerant strain of *E. cloacae* (designated as EC07) and was selected as an effective dose for further experiments.

## Bacterial cell preparation in the treatment of cadmium ions

Two strains of *E. cloacae* designated as EC01 and EC07 were considered according to their heavy metal tolerant ability as tolerant and intolerant strains, respectively. A standard strain of *E. cloacae* (ATCC 13047) was included for comparison. These isolates from frozen stock were initially grown in LB broth at 37 °C for overnight. Then, bacteria were sub-cultured onto LB agar and further incubated at 37 °C for overnight. With three biological replicates, three isolated colonies were inoculated into 50 ml of LB broth and incubated at 37 °C for 18 h. Cells were collected and adjusted to equal cell density at $OD_{600}$ of approximately 4.0 in 50 ml LB broth. Cells were initially grown at 37 °C, 200 rpm for 30 min before treatment with or without 0.2 mM $CdCl_2$. Cells were further incubated for 6 h and harvested by centrifugation at 14,000 rpm for 30 min at 4 °C. Pellets were washed five times with Tris-sucrose buffer (1 mM Tris-base, 250 mM sucrose, pH 7.2) at 14,000 rpm for 30 min at 4 °C to remove the excess salts and heavy metals. The bacterial cells were divided into two parts: the first part was prepared for proteomics analysis and another part for cadmium accumulation analysis, and every step was done on ice. The pellets were stored at −80 °C for proteomics analysis.

## Intracellular cadmium accumulation analysis

For cadmium accumulation analysis, cells from the previous step were adjusted to $OD_{600}$ = 1.0 in 3 ml of Tris-sucrose buffer and were centrifuged at 14,000 rpm, 4 °C for 30 min. Then, the supernatant was removed and 500 μl of concentrated nitric acid was added to the pellet and kept for overnight. The amounts of cadmium accumulation in bacterial cells were measured by atomic absorption spectroscopy (Varian SpectrAA-640; Varian, Palo Alto, CA, USA) and SpectrAA 200 software (Varian, Palo Alto, CA, USA). Briefly, a cadmium stock standard solution (1,000 μg/ml) was diluted to various concentrations with 0.2% $HNO_3$ and served as working standard solutions. Then, the calibration curve (0–1,000 μg/L) was prepared using blank and working standard solutions under the preset parameters (wavelength 228.8 nm with 0.5 nm slit width and the characteristic concentration check (∼0.2 absorbance unit) at 0.5 mg/L). The sample blank contained all reagents used in the sample preparation. Samples were diluted 20 fold prior to cadmium determination and the amount of cadmium was subsequently recorded.

## Proteomics analysis
### Proteins extraction

The bacterial pellets were resuspened in 2× volumes of sample of lysis buffer (2 M thiourea, 7 M urea, 4% CHAPS, and 1% protease inhibitors cocktail). The bacterial cells were disrupted by sonicator (Amplitude 60, 0.5 cycles) on ice. Then, samples were centrifuged at 14,000 rpm for 30 min at 4 °C. Protein concentration was measured by Bradford's method (Bio-Rad protein assay; Bio-Rad Laboratories, Hercules, CA, USA). The protein samples were collected and stored at −80 °C.

### Minimal labeling of protein sample

Protein extracts were initially cleaned-up by using the 2-D Clean-up kit (GE Healthcare, Chicago, IL, USA). Then, the proteins of all samples were labeled with N-hydroxy succinimidyl ester-derivatives of the cyanine dye Cy2, Cy3 and Cy5 (GE Cydye DIGE Fluor (minimal dyes) Labeling Kit (GE Healthcare)) following the manufacturer' protocols. Briefly, the internal standard pool was generated by combining equal 25 μg of extracts from all samples and then the pooled sample was labeled with 400 pmol of Cy2 to aid image matching and cross-gel statistical analysis. The 50 μg proteins from control and cadmium treated cell were minimally labeled with 400 pmol of Cy3 or Cy5. The samples were vortexed and kept on ice for 30 min in the darkness. Then, the reaction was terminated by addition of 1 μl of 10 mM lysine and subsequently incubated on ice for 10 min in the darkness.

### First-dimensional gel electrophoresis

The total of 50 μg protein of each sample, which randomly labeled with Cy3 or Cy5, and the internal standard Cy2, were mixed together and applied to 120 μl of rehydration buffer (8 M urea, 4% CHAPS, 0.001% (w/v) bromophenol blue), 3 mM of dithiothreitol (DTT) and 1.75 μl 1% IPG buffer 3–10 NL). In parallel, 360 μl of rehydration buffer was applied into rehydration tray and strip was placed. Then, the strips were rehydrated in rehydration solution at 20 °C for 16–20 h. The total 150 μg of pooled labeled cy3, cy5 and cy2 samples were focused in a broad range of 18 cm, 3–10 NL of IPG strips (GE healthcare) using cup-loading technique and the isoelectric focusing was carried out using an IPGphor III apparatus (GE Healthcare) according to the following procedures: 500 volts for 500 volt-h, 1,000 volts for 800 volt-h, and 10,000 volts to reach 27,000 volt-h.

### Second-dimensional gel electrophoresis

The 12.5% polyacrylamide gel was prepared at room temperature for 16-18 h for gel polymerization. The strips were equilibrated twice (15 min per each) in equilibrium buffer (50 mM Tris pH 8.8, 6 M urea, 30% glycerol, 2% SDS, 0.03% bromophenol blue) supplemented with 65 mM DTT or 135 mM iodoacetamide. Then, the equilibrated IPG strips were laid onto the surface of a vertical 12.5% polyacrylamide gel and covered with 0.5% agarose gel and 10% bromophenol blue. Separation of protein was carried out at 2.5 watt per gel at 20 °C for 30 min following by 10 watt per gel at 20 °C until the bromophenol blue dye front reached 0.5 cm from the bottom of the gel.

### Image analysis

2D-DIGE gel images were acquired using Typhoon TRIO Variable Mode Imager (GE Healthcare). Differential analysis was performed by ImageMaster 2D Platinum version 7.0 (GE Healthcare) software tool. These included spot intensity calibration, spot detection and background subtraction. Quantification of intensity of each spot was performed in terms of spot volume (area × intensity). The total spot volume normalization method was applied in which the percentage of each spot volume on a gel image is calculated relative to the total volume of all spots on that image. Then, determination of differentially expressed

proteins was conducted by comparing the ratio of % volume values with control and kept at 4 °C until protein identification.

### Gel staining

Gels were stained with Colloidal coomassie brilliant blue staining (10% (w/v) ammonium sulfate, 1% (v/v) phosphoric acid, 0.1% (w/v) Coomassie blue) for overnight. Finally, the gels were destained with milli-Q water until a clear background was observed.

### Protein identification

The QTOF-LC-MS/MS system consisting of a liquid chromatography part (Ultimate 3000, thermo Scientific) in combination with an electrospray ionization (ESI) Quadrupole time of flight mass spectrometer (Model Amazon SL; Bruker, Hamburg, Germany) was used for protein identification. Protein spots of interest were digested by trypsin. First, the protein spots were excised from 2-DE gel and washed twice with 50 μl of 50% acetronitrile (ACN)/25 mM ammonium bicarbonate ($NH_4HCO_3$) at room temperature for 15 min. Then, the solvent was removed and 50 μl of 100% acetonitrile was added for 10 min or until all gels were in white. After that, the acetonitrile was removed and 5 μl of diluted trypsin (diluted 0.1 mg/ml stock trypsin 1:10 into 25 mM ammonium bicarbonate) was added into each tube at 37 °C for 16–24 h. Next, the supernatants were removed to new tubes. Peptides were extracted twice by 15–25 μl of 50% acetonitrile, 5% trifluoroacetic acid (TFA) to each tube containing gel slice for 15 min. Then, the extracted peptides were removed and combined with the supernatant in the new tube. Finally, the extracted peptides were dried in a vacuum centrifuge to dryness. For sample analysis by QTOF-LC-MS/MS, peptides were separated on a reversed phase column (Hypersil GOLD 50 × 0.5 mm, 5 μm C18) and eluted at a flow rate of 100 μl/min under gradient condition of 5–80% B over 50 min mobile phase A consists of water/formic acid (99.9:0.1, v/v) and B consists of acetonitrile/water/formic acid (80:20:0.08, v/v). Mass spectral data from 300 to 1,500 m/z were collected in the positive ionization mode. To identify the protein, all MS/MS spectra recorded on tryptic peptides derived from spots were searched against protein sequences from NCBInr databases using the MASCOT search engine program (available at http://www.matrixscience.com). The functional analysis of protein was performed on UniProtKB database (https://www.uniprot.org/help/uniprotkb).

## Unknown protein searching

The amino acid sequences of unknown protein were searched on BLAST (https://blast.ncbi.nlm.nih.gov/Blast.cgi) using 'non-redundant protein sequences (nr)' database. Then, the secondary structures of unknown protein were predicted by PSIPRED 'server:bioinf.cs.ucl.ac.uk/psipred/'. At last, the tertiary structure was predicted by SWISS-MODEL server (https://swissmodel.expasy.org/interactive).

## Protein network analysis

Protein association networks were created by subjecting the identified proteins to the STRING software version 9.05 (http://string-db.org/). Direct (physical) and indirect (functional) associations were weighted and integrated by using various data, e.g., genetic

context, high-throughput experiments, co-expression and previous knowledge, from the database.

## Statistical analysis

All data are expressed as mean $\pm$ standard deviation (SD). For group comparisons, data were determined by analysis of variance (ANOVA). The strength of each association is presented as the regression coefficient with 95% confidence interval and $P$-value. A $P$-value of <0.05 was considered statistically significant.

## Confirmation of gene expression by qRT-PCR analysis
### RNA isolation and quantification

Bacterial cell suspension was centrifuged for cells harvesting. A 0.75 ml of TRIzol[TM] reagent (Thermo Fisher Scientific, Waltham, MA, USA) was added to $1 \times 10^7$ cells of EC. Cells were lysed by pipetting-up and -down several times. The homogenized samples were transferred to 1.5 ml microcentrifuge tube and further incubated for 5 min at room temperature for completing the dissociation of the nucleoprotein complex. Then, 0.2 ml chloroform (Sigma-Aldrich Chemical Co., St. Louis, MO, USA) was added. Tubes were further mixed vigorously by hand for 15 s and incubated for 2 min at room temperature. After that, the samples were centrifuged at 14,000 rpm for 15 min at 4 °C. The mixture was separated into a lower red phenol-chloroform phase, an interphase, and a colorless upper aqueous phase. Since the RNA remained in the aqueous phase so the aqueous phase of the sample was removed by angling the tube 45° and pipetting the solution out to new centrifuge tube. For RNA precipitation, 0.5 ml of isopropanol (Sigma-Aldrich Chemical Co.) was added to the aqueous phase tubes and incubated at room temperature for 10 min. Tube was centrifuged at 14,000 rpm for 10 min at 4 °C. Finally, the RNA forms a gel-like pellet on the side and bottom of the tube. After precipitation and centrifugation, the supernatant was removed, leaving only RNA pellet. The RNA pellet was washed with 75% ethanol and vortexed briefly. Then, it was centrifuged at 9,000 rpm for 5 min at 4 °C and discarded the wash. The RNA pellet was dried by air for 5–10 min. To quantify the RNA concentration, the RNA pellet was dissolved in 20 µl DEPC-treated water. RNA concentration was quantified with the Nano-Drop Spectrophotometer. RNA quality was evaluated by the absorbance 260/280 nm and 260/230 nm ratios with value >1.8 and >2.0, respectively.

### cDNA synthesis

RNA was reverse transcribed into cDNA using SuperScript® III reverse transcriptase (Thermo Fisher Scientific). The reverse transcription-polymerase chain reaction (RT-PCR) was performed using Mastercycler nexus X2 (Eppendorf, USA). Firstly, each reagent was mixed and briefly centrifuged before use. Starting material of cDNA synthesis was prepared from 5 µg of total RNA, 1 µl of 50 µM oligo(dT)$_{20}$ primer, 1 µl of 10 mM dNTP mix and DEPC-treated water was added until total volume to 10 µl in 0.5 ml tube. Then, the tube was incubated at 65 °C for 5 min and placed on ice for at least 1 min. After that, cDNA synthesis was prepared by mixing of 2 µl of 10× RT buffer, 4 µl of 25 mM MgCl$_2$, 2 µl of 0.1 M DTT, 1 µl of RNaseOUT[TM] (40 U/µl) and 1 µl of SuperScript® III RT (200 U/µl). 10 µl of cDNA synthesis mix was added to RNA-primer mixture, mixed gently, and

incubated at 50 °C for 50 min. The reaction was terminated at 85 °C for 5 min and chilled on ice. Finally, 1 μl of RNaseH was added to each tube and incubated at 37 °C for 20 min to hydrolyze specifically the phosphodiester bonds of RNA hybridized to DNA. cDNA was stored at −20 °C until use for PCR. cDNA concentrations were measured following the RT reaction with the Nano-Drop Spectrophotometer.

***Quantitative reverse transcription-polymerase chain reaction (qRT-PCR)***
The resulting cDNA was used in a 20 μl reaction that contained 1 μl cDNA, 10 μl SYBR® Green mix (Bio-Rad Laboratories, Hercules, CA, USA), 0.4 μl of specific primer sets, the *DUF326* forward: 5′-GTTTGCGCTGATGTGTGTCA-3′ and reverse: 5′-GCCATCCTTCGGCATTGTTC- 3′; the *GAPDH* forward: 5′-CAGATGTCATTGGCTCGCAC- 3′ and reverse: 5′-AGTGCGGCGTATTTATCAAGCG-3′. Then, DEPC-treated water was added until the total volume of 20 μl. Parameters for PCR amplification were as 95 °C for 3 min for enzyme activation, followed by 40 cycles each consisting of denaturation at 95 °C for 3 s, annealing at 54 °C (*DUF326*) and 59 °C (*GAPDH*) for 30 s and extension at 72 °C for 20 s. The *GAPDH* was used as reference gene to normalize values. Relative quantification calculations were performed by $\Delta\Delta Cq$ value.

# RESULTS

## Bacterial identification and minimum inhibitory concentrations (MICs) of cadmium ions in *Enterobacter cloacae* isolated from water and soil

All nine isolates were identified by conventional biochemical identification and further confirmed as *Enterobacter cloacae* with the high scores of 2.013–2.255 by MALDI-Biotyper (Table 1). The MICs of cadmium ions in these isolates were mostly in the range of 0.8–1.6 mM (Table 2). There were only two isolates, namely EC01 and EC08, which exhibited the MICs at 1.6 mM. However, it seems that the EC01 could tolerate the other metal ions (e.g., zinc and copper) approximately two-fold higher than those of the EC08 strain. The same degree of tolerance against these three kinds of metal ions was also found in the standard strain (ATCC 13047). To further elucidate the cellular responses and the underlying mechanism against cadmium ions, the EC07 (isolated from the cleaned area) was selected since it showed 4 fold lower in MIC than that of the EC01 strain while displaying the same degree of tolerance against zinc and copper ions. Therefore, in the present study, three strains of *E. cloacae* (EC01, EC07 and ATCC 13047) were selected for further experiments.

## Growth patterns of *E. cloacae* isolates in the presence of sub-lethal dose of cadmium ions

To investigate the growth patterns of these three strains in the presence of cadmium ions, the final concentration of 0.2 mM CdCl$_2$ represented as MIC/2 of the most sensitive strain (EC07) was selected. As shown in Fig. 1B, some suppressing effect on the growth arrest of EC01 was found at the lag phase and early-log phase while the rate of cell division at the middle- to late-log phase resembled those in the absence of cadmium ions. Such recovery was not significantly detected in the case of ATCC 13047 (Fig. 1A). Not surprisingly,

**Table 1** Genus and species identification of nine environmental isolates and ATCC strain of *E. cloacae* by MALDI-TOF mass spectrometry.

| No. | Name | MALDI-TOF/MS | | | |
|---|---|---|---|---|---|
| | | Organism (best match) | Score value | Organism (second best match) | Score value |
| 1 | *E. cloacae* 01 | *Enterobacter cloacae* | 2.013 | *Enterobacter cloacae* | 2.002 |
| 2 | *E. cloacae* 02 | *Enterobacter cloacae* | 2.122 | *Enterobacter kobei* | 2.099 |
| 3 | *E. cloacae* 05 | *Enterobacter cloacae* | 2.072 | *Enterobacter cloacae* | 2.011 |
| 4 | *E. cloacae* 06 | *Enterobacter cloacae* | 2.177 | *Enterobacter cloacae* | 2.004 |
| 5 | *E. cloacae* 07 | *Enterobacter cloacae* | 2.255 | *Enterobacter cloacae* | 2.172 |
| 6 | *E. cloacae* 08 | *Enterobacter cloacae* | 2.142 | *Enterobacter cloacae* | 2.068 |
| 7 | *E. cloacae* 09 | *Enterobacter cloacae* | 2.173 | *Enterobacter cloacae* | 2.046 |
| 8 | *E. cloacae* 10 | *Enterobacter cloacae* | 2.099 | *Enterobacter cloacae* | 1.997 |
| 9 | *E. cloacae* 11 | *Enterobacter cloacae* | 2.08 | *Enterobacter kobei* | 2.066 |
| 10 | ATCC 13047 | *Enterobacter cloacae* | 2.001 | *Enterobacter cloacae* | 1.965 |

**Notes.**

Meaning of score values derived from MALDI/TOF mass spectrometry are as follows:

-Values in the range of 2.300–3.000 denote the highly probable species identification (represented in green).

-Values in the range of 2.000–2.299 denote the secure genus identification, probable species identification (represented in green).

-Values in the range of 1.700–1.999 denote the secure genus identification (represented in yellow).

-Values in the range of 0.000–1.699 denote the unreliable identification (represented in red).

**Table 2** The minimum inhibitory concentrations (MICs) of cadmium, zinc and copper ions of nine environmental strains and ATCC 13047 strain of *E. cloacae*.

| No. | Name | Minimal Inhibitory Concentration (MIC) test | | |
|---|---|---|---|---|
| | | $Cd^{2+}$ (mM) | $Zn^{2+}$ (mM) | $Cu^{2+}$ (mM) |
| 1 | ATCC 13047 | 1.6 | 3.2 | 8 |
| 2 | *E. cloacae* 01 | 1.6 | 3.2 | 8 |
| 3 | *E. cloacae* 02 | 0.8 | 1.6 | 8 |
| 4 | *E. cloacae* 05 | 0.8 | 1.6 | 8 |
| 5 | *E. cloacae* 06 | 0.8 | 1.6 | 4 |
| 6 | *E. cloacae* 07 | 0.4 | 3.2 | 8 |
| 7 | *E. cloacae* 08 | 1.6 | 1.6 | 4 |
| 8 | *E. cloacae* 09 | 0.8 | 1.6 | 8 |
| 9 | *E. cloacae* 10 | 0.8 | 1.6 | 8 |
| 10 | *E. cloacae* 11 | 0.8 | 1.6 | 4 |

cadmium ions at 0.2 mM exerted their toxic effects on the growth of EC07 at all phases of cell division (Fig. 1C).

## Cadmium accumulation at the cellular and protein levels

Next question was addressed on whether the tolerance phenomenon found in EC01 associated with the higher cadmium accumulation intracellularly. As illustrated in Fig. 2, the higher cadmium adsorptivity of upto 2.7–3.0 µg per $8 \times 10^8$ cells was found in the case of EC01, particularly at 6 h, as compared to the others. Importantly, when quantification of cadmium adsorption was performed in the whole cell lysates, the proteins portion of EC01 provided binding capability of approximately two-fold higher than those of the ATCC

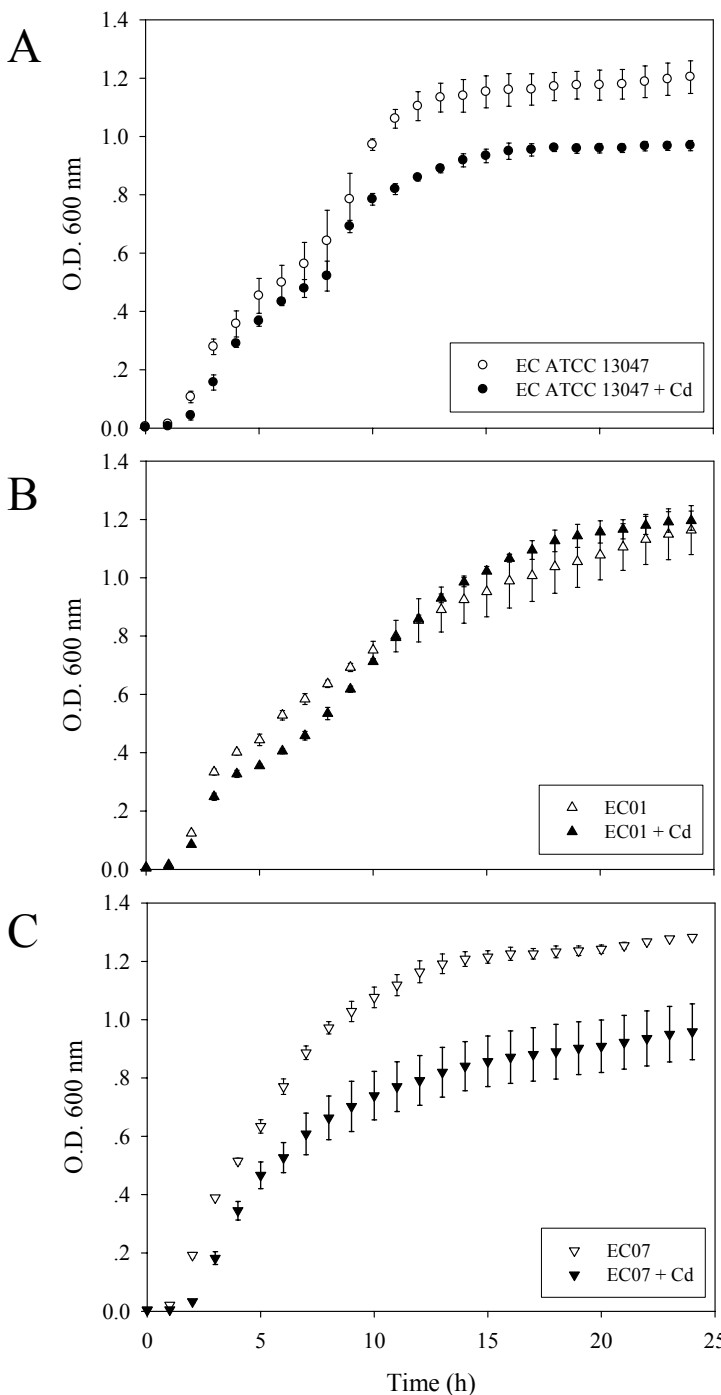

**Figure 1** **Growth curves of *E. cloacae* strains ATCC 13047 (A), EC01 (B) and EC07 (C) in the absence (opened symbol) and presence (closed symbol) of 0.2 mM CdCl$_2$.** Optical density at 600 nm was monitored at 1-h intervals for 24 h.

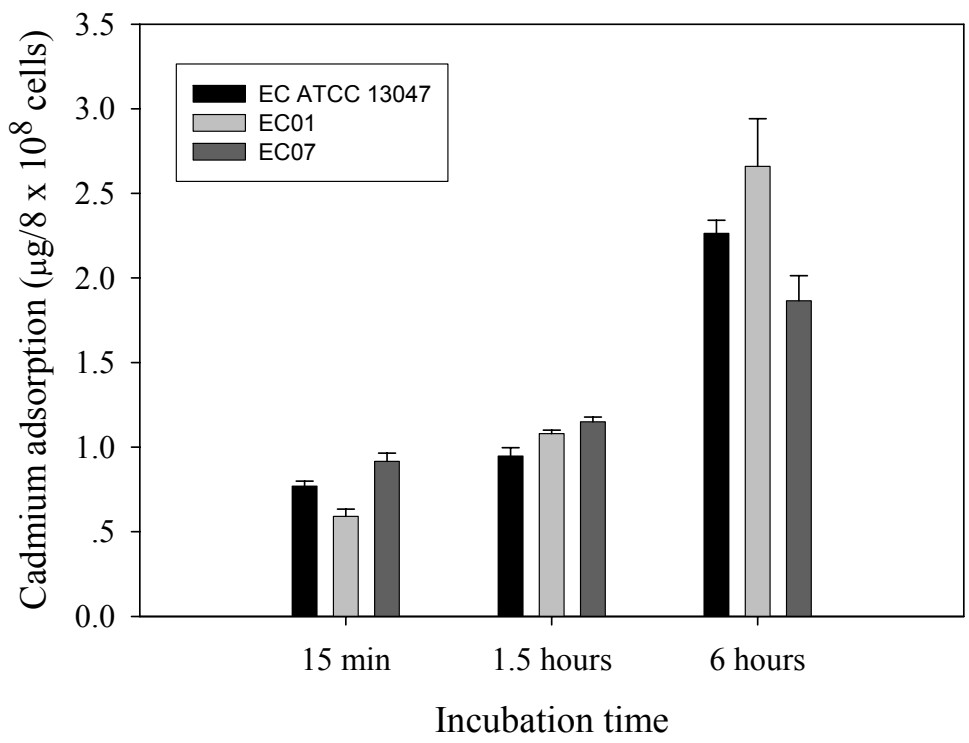

**Figure 2** **Time-course cadmium adsorption by *E. cloacae* tolerant strain (EC01), intolerant strain (EC07) and ATCC strain grown in LB broth containing 0.2 mM CdCl₂ at 37 °C.** Error bars represent as mean ± SD.

13047 and EC07 strains (Fig. 3). Our findings lend support to the hypothesis that the EC01 expressed some of the cadmium-responsive proteins differently from those of the others.

## Protein expression profiles of *E. cloacae* in the presence of cadmium ions revealed by 2D-DIGE

To further explore the detailed mechanism on how *E. cloacae* initially responded and adopt themselves upon exposure to cadmium toxicity in a short period of time (6 h), two-dimensional-difference in gel electrophoresis (2D-DIGE) in conjuction with protein identification *via* mass spectrometry was utilized. As demonstrated in Fig. 4, approximately 250 protein spots were detected in the master map of pooled proteins (labeled with Cy2) served as an internal standard for these three isolates. Eighty protein spots were then picked up and analyzed by the QTOF-LC-MS/MS system for protein identification (Table 3). Most of the proteins were classified by UniProtKB to be responsive for oxidative stress, protein folding, glycolytic process, translation, metal ion binding, ion transport and homeostasis (Fig. 5). Up- or down-regulations of protein in *E. cloacae* ATCC 13047, EC01 and EC07 strains were summarised in Table 3. In case of ATCC 13047, up-regulation of catalase (spots 2 and 4), Hsp20 (spot 28), malate dehydrogenase (spot 32), glyceraldehyde-3-phosphate dehydrogenase (spots 36 and 37), superoxide dismutase (spot 38), fructose-bisphosphate aldolase (spot 46), GroEL (spots 55, 56 and 59), putative hydrolase (spot 58), ATP synthase

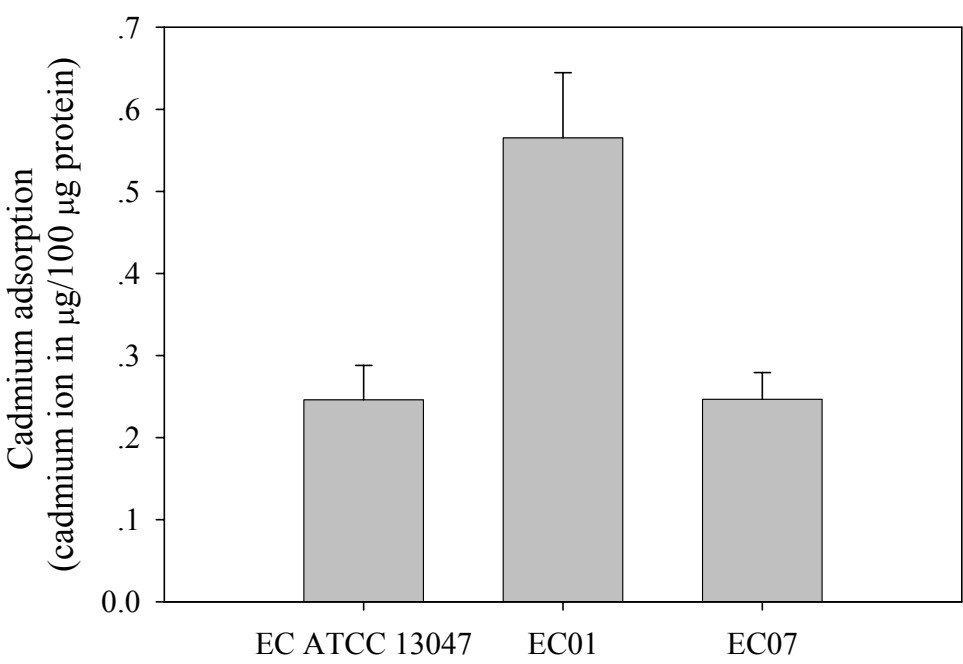

**Figure 3** **Cadmium adsorptivity of whole cell proteins from *E. cloacae* tolerant strain (EC01), intolerant strain (EC07) and ATCC strain.** Error bars represent as mean ± SD.

(spot 67) was detected in response to cadmium toxicity. For the EC01, a set of protein similar to those of the ATCC strain was up-regulated such as catalase (spots 1, 2 and 3), Hsp20 (spot 28), malate dehydrogenase (spot 32), glyceraldehyde-3-phosphate dehydrogenase (spots 36 and 37), fructose-bisphosphate aldolase (spot 46) and GroEL (spots 55, 56 and 59). Meanwhile, up-regulation of some proteins including dihydrolipoamide dehydrogenase (spot 10), enolase (spot 52), domain of uncharacterized function (DUF 326) (spot 57), aspartate ammonia-lyase (spot 60) and glutamine-binding periplasmic protein (spot 79) was found in the EC01. The protein profile of EC07 showed the up-regulation of catalase (spot 3), hydroperoxidase (spot 11), Dps (spot 19), OmpX (spot 23), YciF protein (spot 27), TrpR binding protein WrbA (spot 29), OmpC porin (spot 45), glucose-specific PTS system (spot 68) and glutamine-binding periplasmic protein (spot 79).

To further simplify the groups of protein shown as unique or overlap proteins among these three strains of *E. cloacae*, Venn diagram was plotted as illustrated in Fig. 6. Catalase was analyzed to be the only one protein found in all three strains under cadmium stress. Four proteins namely enolase, DUF326-like domain, dihydrolipoamide dehydrogenase, aspartate ammonia-lyase were detected as unique proteins in the case of tolerant strain (EC01). In the intolerant strain (EC07), the unique proteins were identified as OmpX, protein YciF, OmpC porin, DNA protection during starvation protein, and TrpR binding protein WrbA. However, our finding revealed that glutamine-binding periplasmic protein was the only one protein shared between EC01 and EC07 strains.

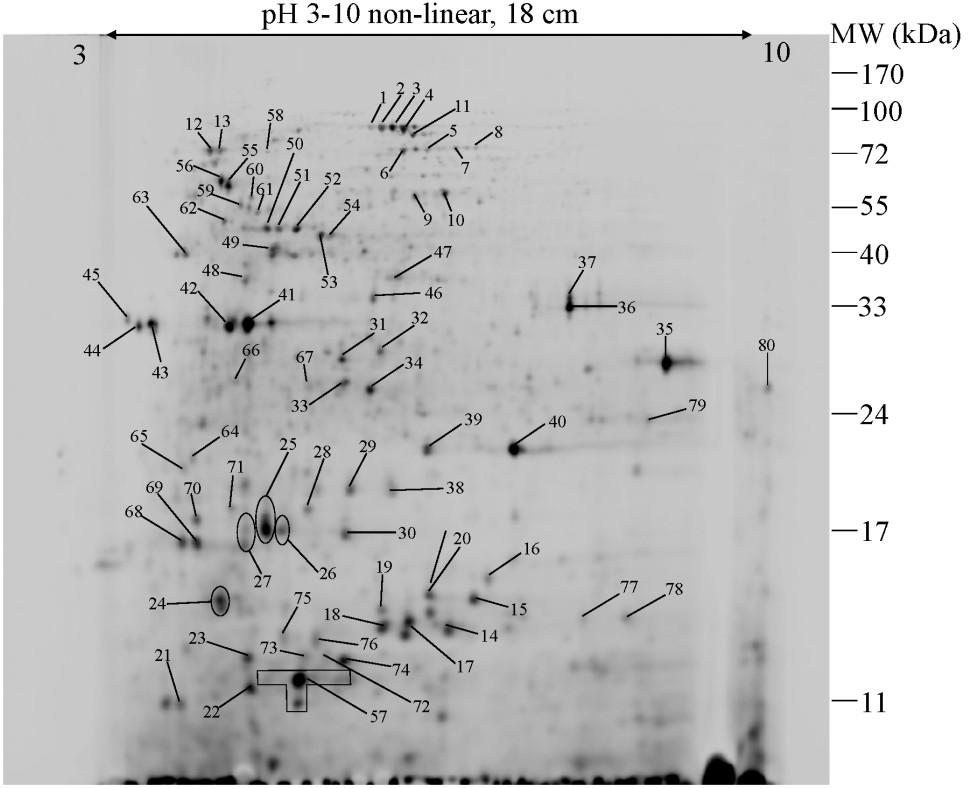

**Figure 4  2-DE image of 350 μg of total soluble proteins extracted from *E. cloacae* ATCC 13047.** The proteins were immobilized on Immobiline Dry strips with a nonlinear pH gradient from 3 to 10 followed by SDS-PAGE on 12.5% polyacrylamide gels. Protein spots were visualized by coomassie brilliant blue R-250 (CBB) staining. Eighty protein spots were identified as marked with arrows.

## Identification of specific cadmium-induced protein alterations in EC01 (tolerant strain) and EC07 (intolerant strain)

Differentially expressed proteins found in EC01 as compared to those of the EC07 strain in the presence of cadmium ions were analyzed as shown in Tables 4 and 5. The DUF326-like domain was analyzed to show the highest difference in fold change (+220.92 fold) of protein expression between EC01 and EC07 (Table 4). This indicates that the EC01 normally expresses the DUF326-like domain at high level when grows up in the environmental-contaminated site. Therefore, exposure to cadmium ions at 0.2 mM in a short period of time (6 h) triggered the cells to up-regulate the DUF326-like domain expression of only 1.35 fold higher than that in the absence of cadmium. Similar phenomena were also found in the regulation of antioxidative enzymes such as hydroperoxidase II (+42.3 and +21.3 fold) and superoxide dismutase (+8.14 fold), expression of periplasmic protein (+14.3 fold) or outer membrane protein X (+6.98 fold) as well as the metabolic enzyme (MDH; +7.45 fold) as compared to very low/undetectable amount found in the control. Beside these, differences in expression of the other proteins were found in the range of 1.38–6 fold higher than those of the EC07 in response to the toxic cadmium ions.

Peer*J*

**Table 3  Differentially expressed proteins of *E. cloacae* represented as fold changes in response to cadmium treatment.** Proteins were identified by QTOF mass spectrometry.

| Spot no. | Accession no. | Protein names | Fold change ATCC 13047 + Cd | Fold change EC01 + Cd | Fold change EC07 + Cd | Anova | M.W. | pI | Score | % protein seq. coverage | Matched peptide |
|---|---|---|---|---|---|---|---|---|---|---|---|
| 1 | KGB06270.1 \| | Catalase HPII | −1.07 | 1.76 | −1.11 | 3.42E−03 | 83,514 | 5.63 | 331 | 14 | 28(14) |
| 2 | KGB06270.1 \| | Catalase HPII | 2.23 | 2.33 | −1.04 | 2.89E−04 | 83,514 | 5.63 | 314 | 7 | 13(9) |
| 3 | KGB06270.1 \| | Catalase HPII | −1.15 | 1.34 | 1.31 | 5.33E−01 | 83,514 | 5.63 | 1,587 | 16 | 79(67) |
| 4 | KGB06270.1 \| | Catalase HPII | 1.57 | 1.14 | 1.21 | 1.02E−05 | 83,514 | 5.63 | 1,396 | 22 | 76(63) |
| 5 | KJC00120.1 \| | Hydroperoxidase | −1.56 | −1.59 | −2.04 | 1.70E−07 | 83,488 | 5.66 | 298 | 13 | 24(14) |
| 6 | CZY07068.1 \| | Hydroperoxidase II | Undetectable | −1.85 | Undetectable | 4.33E−02 | 83,493 | 5.76 | 151 | 7 | 9(7) |
| 7 | SAG51425.1 \| | Hydroperoxidase II | Undetectable | −1.06 | Undetectable | 1.52E−02 | 83,473 | 5.61 | 1,471 | 24 | 85(71) |
| 8 | KJC00120.1 \| | Hydroperoxidase II | Undetectable | Undetectable | −1.87 | 2.82E−02 | 83,488 | 5.66 | 791 | 18 | 43(35) |
| 9 | KJX08087.1 \| | Succinate-semialdehyde dehydrogenase | 1.22 | −1.06 | −1.13 | 7.38E−10 | 49,930 | 5.71 | 516 | 25 | 89(34) |
| 10 | KTI00918.1 \| | Dihydrolipoamide dehydrogenase | −1.03 | 1.46 | −1.05 | 1.66E−06 | 50,867 | 5.79 | 302 | 16 | 12(10) |
| 11 | KJC00120.1 \| | Hydroperoxidase | Undetectable | Undetectable | 3.88 | 3.11E−04 | 83,488 | 5.66 | 580 | 20 | 34(27) |
| 12 | ADF60390.1 \| | Chaperone protein DnaK | −1.38 | −1.06 | −1.98 | 1.39E−01 | 69,092 | 4.82 | 120 | 13 | 13(8) |
| 13 | SAB00249.1 \| | 30S ribosomal protein S1 | −2.14 | −1.28 | −2.04 | 1.40E−01 | 61,289 | 4.89 | 134 | 14 | 23(8) |
| 14 | KLG03309.1 \| | Peroxiredoxin OsmC | −7.07 | −2.44 | −1.15 | 6.58E−03 | 15,005 | 6.04 | 217 | 52 | 52(21) |
| 15 | KGB03490.1 \| | DNA protection during starvation protein | −7.51 | −3.74 | −1.01 | 1.39E−02 | 18,720 | 5.94 | 1,022 | 76 | 205(84) |
| 16 | KPU05892.1 \| | Superoxide dismutase | 1.26 | −1.16 | −1.23 | 2.94E−02 | 17,768 | 6.43 | 766 | 33 | 66(33) |
| 17 | ADF61100.1 \| | Type VI secretion system secreted protein Hcp | Not analysis | Not analysis | Not analysis | Not analysis | 16,996 | 5.76 | 432 | 40 | 25(17) |
| 18 | SAH00153.1 \| | OsmC family protein | Undetectable | Undetectable | −1.91 | 5.04E−08 | 15,149 | 5.74 | 156 | 52 | 18(13) |
| 19 | SAG00330.1 \| | DNA protection during starvation protein | Undetectable | Undetectable | 2.01 | 4.55E−05 | 18,677 | 5.72 | 819 | 36 | 126(60) |
| 20 | SAG00330.1 \| | DNA protection during starvation protein | Undetectable | Undetectable | −5.1 | 4.68E−07 | 18,677 | 5.72 | 506 | 38 | 24(21) |
| 21 | KJQ00352.1 | 50S ribosomal protein L21 | −1.33 | −1.09 | 1.1 | 6.97E−06 | 11,543 | 9.85 | 58 | 7 | 1(1) |
| 22 | EUM00092.1 \| | Universal stress protein A | −1.84 | −1.15 | −1.38 | 7.28E−05 | 16,200 | 4.98 | 53 | 23 | 4(3) |
| 23 | AQT90079.1 \| | Outer membrane protein OmpX | −1.28 | −1.22 | 1.3 | 1.35E−07 | 18,654 | 6.23 | 1,064 | 22 | 59(36) |
| 24 | SAF00184.1 \| | Protein YciE | Undetectable | Undetectable | −1.74 | 2.68E−04 | 19,036 | 4.85 | 173 | 28 | 12(10) |
| 25 | SAF00211.1 \| | Protein YciF | Undetectable | Undetectable | −1.06 | 4.32E−09 | 18,313 | 5.04 | 1,838 | 29 | 187(130) |

**Table 3** (*continued*)

| Spot no. | Accession no. | Protein names | Fold change ATCC 13047 + Cd | Fold change EC01 + Cd | Fold change EC07 + Cd | Anova | M.W. | pI | Score | % protein seq. coverage | Matched peptide |
|---|---|---|---|---|---|---|---|---|---|---|---|
| 26 | OUF00075.1 \| | Osmotically-inducible protein Y | Not analysis | Not analysis | 1.1 | 3.72E−03 | 21,426 | 6.17 | 1,181 | 35 | 53(44) |
| 27 | CZX11689.1 \| | Protein YciF | Undetectable | Undetectable | 2.45 | 3.65E−05 | 18,325 | 4.95 | 535 | 60 | 138(35) |
| 28 | KPU03204.1 \| | Heat-shock protein Hsp20 | 1.3 | 2.39 | 1.29 | 6.21E−05 | 21,493 | 5.32 | 687 | 48 | 66(36) |
| 29 | SAD00058.1 \| | TrpR binding protein WrbA | −1.31 | −1.01 | 1.41 | 8.72E−10 | 20,847 | 5.78 | 255 | 25 | 14(12) |
| 30 | KJX08128.1 \| | Periplasmic protein | −1.04 | −1.78 | −1.75 | 1.69E−05 | 21,374 | 8.63 | 1,272 | 54 | 73(51) |
| 31 | CZW02624.1 \| | Periplasmic binding protein/LacI transcriptional regulator | −2.05 | 1.011 | −1.34 | 6.04E−09 | 35,617 | 5.88 | 1,576 | 53 | 136(88) |
| 32 | OUF11927.1 \| | Malate dehydrogenase | 1.32 | 1.43 | 1.05 | 1.17E−09 | 32,673 | 5.9 | 1,086 | 42 | 55(42) |
| 33 | CZX00043.1 \| | 2,5-diketo-D-gluconate reductase A | −1.09 | 1.21 | −1.07 | 1.46E−02 | 30,934 | 5.49 | 219 | 54 | 42(19) |
| 34 | SAD00983.1 \| | Cationic amino acid ABC transporter substrate-binding protein | −1.37 | −1.27 | −1.45 | 4.27E−01 | 28,179 | 7.6 | 333 | 23 | 24(16) |
| 35 | KGB04009.1 \| | Bacterial extracellular solute-bindings, 3 family protein | −1.49 | −1.91 | 1.14 | 6.37E−04 | 33,362 | 8.8 | 565 | 42 | 51(28) |
| 36 | KJC00085.1 \| | Glyceraldehyde-3-phosphate dehydrogenase | 1.37 | 1.45 | 1.06 | 1.32E−02 | 35,648 | 6.61 | 585 | 42 | 169(58) |
| 37 | SAJ08363.1 \| | Glyceraldehyde-3-phosphate dehydrogenase | 1.62 | 1.72 | 1.07 | 5.94E−06 | 35,650 | 6.61 | 189 | 17 | 21(19) |
| 38 | ADF61881.1 \| | Superoxide dismutase | 1.59 | −1.05 | 1.17 | 1.01E−02 | 21,457 | 5.54 | 46 | 17 | 119(4) |
| 39 | SAD00058.1 \| | TrpR binding protein WrbA | −1.35 | −1.7 | −1.06 | 5.17E−02 | 20,847 | 5.78 | 1,160 | 35 | 51(40) |
| 40 | KTI00016.1 \| | Superoxide dismutase | −1.08 | −1.05 | 1.19 | 4.00E−01 | 22,995 | 6.23 | 204 | 13 | 21(12) |
| 41 | KTI01411.1 \| | Hypothetical protein ASV14_00320 | −1.21 | −1.17 | 1.17 | 2.53E−07 | 37,677 | 5.33 | 1,254 | 47 | 88(54) |
| 42 | CZW04458.1 \| | Outer membrane protein A | −1.14 | −1.43 | 1.24 | 4.62E−08 | 37,577 | 5.2 | 1,791 | 48 | 122(74) |
| 43 | KJX05875.1 \| | Porin [*Enterobacter cloacae* subsp. *cloacae*] | −1.36 | −1.26 | 1.27 | 5.81E−09 | 39,231 | 4.58 | 493 | 32 | 44(22) |
| 44 | 5FVN \|A | Chain A, X-ray Crystal Structure of *Enterobacter cloacae* Ompe36 Porin | −1.28 | 1.06 | 1.09 | 2.01E−07 | 37,857 | 4.39 | 3,805 | 35 | 155(133) |
| 45 | SAF58867.1 \| | OmpC porin | −1.24 | −2.06 | 1.38 | 8.59E−07 | 39,850 | 4.44 | 573 | 35 | 78(35) |
| 46 | KJQ00573.1 \| | Fructose-bisphosphate aldolase | 1.5 | 1.6 | −1.18 | 3.50E−03 | 39,384 | 5.52 | 124 | 23 | 14(4) |
| 47 | SAH00347.1 \| | Alcohol dehydrogenase | −1.25 | 1 | −1.15 | 2.21E−05 | 35,832 | 5.7 | 1,725 | 60 | 231(109) |
| 48 | KPU01695.1 \| | Elongation factor Tu | −1.57 | −1.55 | 1.12 | 2.09E−03 | 43,460 | 5.25 | 494 | 27 | 101(32) |
| 49 | KPU01695.1 \| | Elongation factor Tu | −1.59 | −1.17 | −1.46 | 4.33E−02 | 43,460 | 5.25 | 481 | 29 | 28(19) |
| 50 | CZW60501.1 \| | Isocitrate dehydrogenase | −1.36 | 1.17 | −1.25 | 9.02E−03 | 46,145 | 5.14 | 127 | 23 | 17(8) |
| 51 | CZW60501.1 \| | Isocitrate dehydrogenase | −1.18 | 1.18 | −1.51 | 1.49E−08 | 46,145 | 5.14 | 79 | 9 | 4(3) |
| 52 | SAH00247.1 \| | Enolase | 1.19 | 1.56 | −1.35 | 1.65E−03 | 45,632 | 5.19 | 85 | 10 | 13(6) |
| 53 | SAI00778.1 \| | Elongation factor Tu | −1.39 | 1.25 | −1.8 | 3.38E−02 | 43,444 | 5.25 | 673 | 33 | 50(32) |
| 54 | WP_075208257.1 \| | Translation elongation factor Tu | −1.37 | 1.04 | −1.84 | 2.35E−05 | 40,036 | 5.01 | 495 | 40 | 36(24) |

**Table 3** (*continued*)

| Spot no. | Accession no. | Protein names | Fold change ATCC 13047 + Cd | Fold change EC01 + Cd | Fold change EC07 + Cd | Anova | M.W. | pI | Score | % protein seq. coverage | Matched peptide |
|---|---|---|---|---|---|---|---|---|---|---|---|
| 55 | KJJ00122.1 \| | Molecular chaperone GroEL | 2.29 | 2.51 | −1.29 | 7.48E−06 | 57,205 | 4.88 | 800 | 38 | 48(40) |
| 56 | KPU06091.1 \| | Molecular chaperone GroEL | 1.32 | 1.33 | −1.09 | 3.85E−06 | 57,209 | 4.85 | 484 | 36 | 43(32) |
| 57 | SAB11354.1 \| | Domain of Uncharacterised Function (DUF326) | Undetectable | 1.35 | Undetectable | 9.38E−11 | 13,306 | 5.27 | 3,402 | 66 | 123(110) |
| 58 | CZV11460.1 \| | Putative hydrolase | 1.59 | −1.1 | −1.24 | 1.98E−02 | 72,227 | 5.28 | 213 | 10 | 16(10) |
| 59 | KPU06091.1 \| | Molecular chaperone GroEL | 1.85 | 1.36 | 1.09 | 1.56E−02 | 57,209 | 4.85 | 800 | 27 | 69(58) |
| 60 | KJJ00126.1 \| | Aspartate ammonia-lyase | −1.59 | 1.55 | −1.37 | 9.13E−06 | 52,776 | 5.05 | 140 | 23 | 46(14) |
| 61 | SAB00249.1 \| | 30S ribosomal protein S1 | −4.92 | −2.24 | 1.03 | 2.31E−03 | 61,289 | 4.89 | 139 | 10 | 25(12) |
| 62 | SAJ00217.1 \| | F0F1 ATP synthase subunit beta | 1.29 | 1.29 | −1.14 | 2.46E−04 | 50,268 | 4.89 | 305 | 19 | 113(30) |
| 63 | CZY00919.1 \| | Maltoporin | −1.25 | 1.08 | 1.21 | 1.38E−04 | 48,457 | 4.79 | 471 | 27 | 58(22) |
| 64 | SAB00172.1 \| | Tellurium resistance protein TerE | −1.75 | −1.66 | −1.27 | 1.60E−02 | 20,437 | 4.69 | 688 | 30 | 52(32) |
| 65 | SAB00191.1 \| | Tellurium resistance protein TerD | −1.1 | −1.61 | −1.28 | 3.66E−05 | 20,573 | 4.64 | 1,803 | 36 | 58(57) |
| 66 | KTI00836.1 \| | Molecular chaperone DnaK | −1.73 | −1.61 | 1.22 | 4.85E−04 | 69,092 | 4.82 | 473 | 9 | 43(20) |
| 67 | WP_004144995.1 \| | MULTISPECIES: ATP synthase subunit alpha | 2.98 | 1.24 | −1.19 | 4.42E−03 | 55,363 | 5.73 | 99 | 19 | 23(8) |
| 68 | CZZ00122.1 \| | Glucose-specific PTS system component | Undetectable | Undetectable | 2.16 | 1.38E−01 | 18,313 | 5.04 | 104 | 7 | 3(3) |
| 69 | CZZ00122.1 \| | Glucose-specific PTS system component | −3.03 | −1.71 | −1.61 | 6.63E−05 | 18,222 | 4.73 | 660 | 27 | 32(23) |
| 70 | KGZ01983.1 \| | Chemical-damaging agent resistance protein C note like to terD domain protein and Tellurium resistance protein TerE | −3.64 | −1.5 | 1.02 | 1.71E−01 | 20,463 | 4.69 | 392 | 39 | 56(25) |
| 71 | SAF00152.1 \| | Manganese catalase | Undetectable | Undetectable | 1.13 | 3.21E−05 | 31,431 | 4.8 | 338 | 20 | 41(13) |
| 72 | KJJ00123.1 \| | Molecular chaperone GroES | −1.05 | 1.11 | 1.27 | 1.02E−05 | 10,354 | 5.38 | 83 | 27 | 9(7) |
| 73 | KJJ00123.1 \| | Molecular chaperone GroES | Undetectable | −1.13 | 1.29 | 4.92E−03 | 10,354 | 5.38 | 70 | 14 | 3(3) |
| 74 | KJJ00123.1 \| | Molecular chaperone GroES | −1.21 | −1.32 | −1.25 | 1.44E−01 | 10,354 | 5.38 | 555 | 58 | 71(36) |
| 75 | Q84FI1.1 \|DPS_ENTCL | RecName: Full = DNA protection during starvation protein | Undetectable | Undetectable | −1.21 | 8.29E−07 | 18,691 | 5.95 | 133 | 23 | 10(9) |
| 76 | SAC00082.1 \| | LysM domain/BON superfamily protein | −3.09 | −1.52 | Undetectable | 5.26E−06 | 15,895 | 5.4 | 155 | 24 | 9(9) |
| 77 | KJC00085.1 \| | Glyceraldehyde-3-phosphate dehydrogenase | −3.26 | −2.55 | Undetectable | 3.38E−07 | 35,648 | 6.61 | 227 | 16 | 47(15) |

**Table 3** (*continued*)

| Spot no. | Accession no. | Protein names | Fold change ATCC 13047 + Cd | Fold change EC01 + Cd | Fold change EC07 + Cd | Anova | M.W. | *p*I | Score | % protein seq. coverage | Matched peptide |
|---|---|---|---|---|---|---|---|---|---|---|---|
| 78 | KPU04452.1 \| | Ecotin | −1.08 | −1.56 | Undetectable | 1.79E−08 | 19,034 | 8.35 | 208 | 28 | 22(11) |
| 79 | ESM17978.1 \| | Glutamine-binding periplasmic protein | −1.03 | 2.38 | 1.8 | 1.04E−02 | 26,950 | 8.91 | 130 | 15 | 13(8) |
| 80 | KGB04009.1 \| | Bacterial extracellular solute-bindings, 3 family protein | Not analysis | Not analysis | Not analysis | Not analysis | 33,362 | 8.8 | 884 | 19 | 55(52) |

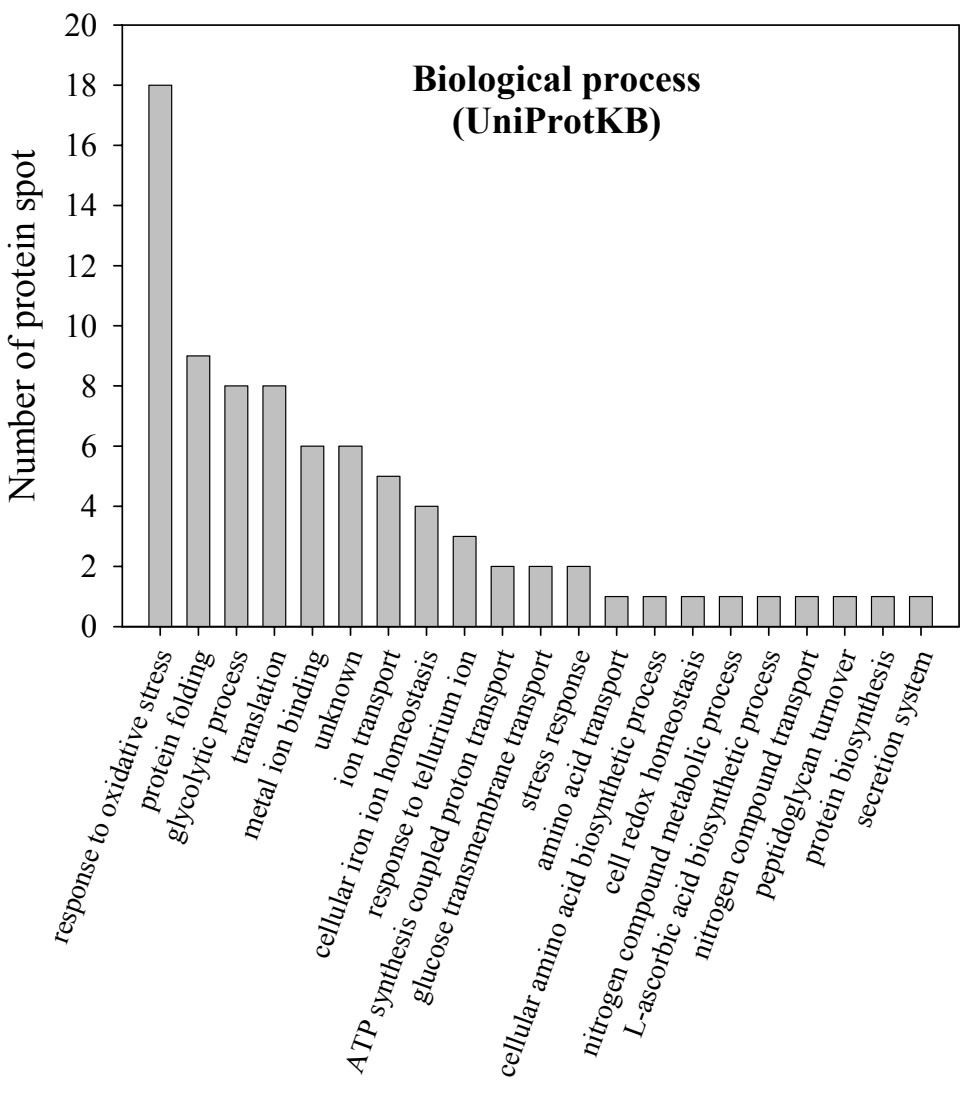

Cellular pathways

**Figure 5** **Numbers of proteins involved in various biological processes of *E. cloacae* ATCC 13047.**
Functional analysis of proteins was identified by UniProtKB database.

As compared to the results of EC07 shown in Table 5, the EC01 exhibited lower expression of YciF protein of approximately 84.7 fold than that of the EC07. Expression of the OmpC porin and manganese catalase of upto 53.2 and 51.7 fold lower than those of the EC07 was also detected. More than 30 fold difference in the osmotically-inducible protein Y and Dps were recorded. Expression of the other proteins was varied in the range of 1.3–29.2 fold.

To further confirm whether cadmium stress affected gene expression of *E. cloacae*, Fig. 7 depicts qRT-PCR analysis of DUF326-like domain expression. Data are normalized relative to GAPDH gene (served as a housekeeping gene) and expressed as the fold difference

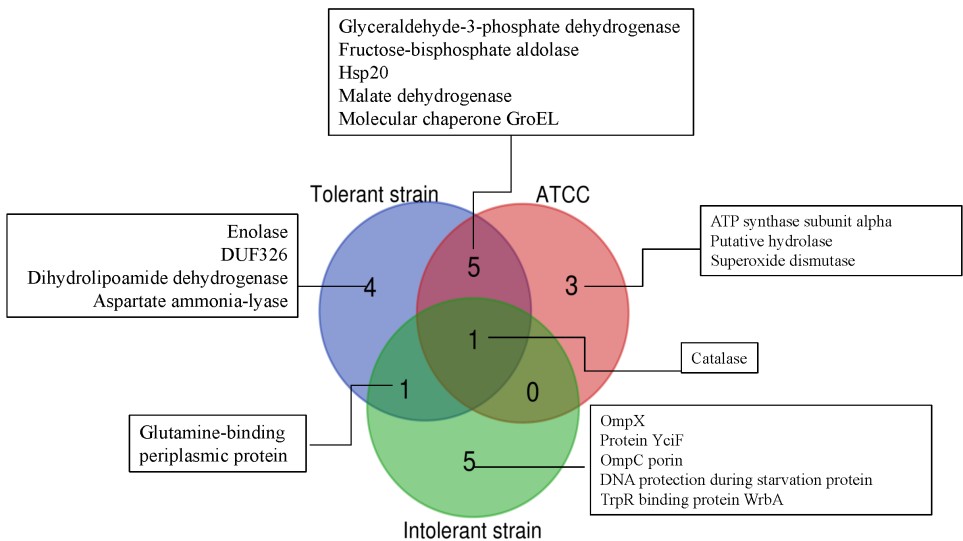

**Figure 6** Venn diagram showing the number of significantly up-regulated proteins from 3 strains of *E. cloacae* with *P*-value ≤ 0.05 and fold change ≥ 1.3 under cadmium stress. The diagram indicates the overlap between the tolerant (EC01), intolerant (EC07) and ATCC 13047 strains.

($2^{-\Delta\Delta Cq}$) between control and cadmium-treated group. Expression of DUF326 gene in *E.cloacae* strain 01, 07 and ATCC were at 14.13, 1.28, 1.49-fold higher in cadmium-treated than in the control group, respectively.

## DISCUSSION

Cadmium contamination remains a major health concern due to its toxicity to human, animals and microorganisms. Searching for bacteria that possess high capability to bind cadmium ions and/or cadmium tolerance is needed for further applying as bioremediation tool. *E. cloacae* has been isolated and characterized as a potential heavy metal accumulating bacteria (*Iyer, Mody & Jha, 2004*; *Naik, Pandey & Dubey, 2012*; *Rahman et al., 2015*). However, the global-protein response to cadmium of *E. cloacae* is still unclear. In this study, the cadmium-tolerant *E. cloacae* isolated from agricultural and industrial area were collected. Minimum inhibitory concentrations (MICs) and growth patterns in the presence of cadmium ions have been tested. As expected, the tolerant strain (EC01) with the MIC of 1.6 mM cadmium has been isolated from industrial effluents of heavy metal area whereas the intolerant strain (EC07) (MIC at 0.4 mM) has been isolated from the cleaned area. Even though, the ATCC 13047 exhibited the same MIC as that of the EC01, recovery of growth pattern during exposure to cadmium was in a lesser extent than the EC01. Moreover, the degree of cadmium binding by the whole cell lysate of EC01 was approximately 2 fold higher than those of the ATCC 13047 and EC07. Such cadmium binding property opens up a high feasibility of utilizing as biosorbents for cadmium remediation as in case of the exopolysaccharides (*Iyer, Mody & Jha, 2005*).

To explore the underlying mechanisms of *E. cloacae* in response to cadmium toxicity, proteomics profiling of these three strains were carried out by 2D-DIGE technology.

**Table 4  Regulation of major responsive proteins found in EC01 (cut-off at 1.3 fold).**

| Spot no. | Accession no. | Protein names | Ratio of 01-Cd/07-Cd | Ratio of 01-Cd/01-control | Anova |
|---|---|---|---|---|---|
| 57 | SAB11354.1 \| | Domain of Uncharacterized Function (DUF326) | +220.92 | +1.35 | 9.38E−11 |
| 6 | CZY07068.1 \| | Hydroperoxidase II | +42.29 | U.D. | 4.33E−02 |
| 7 | SAG51425.1 \| | Hydroperoxidase II | +21.28 | U.D. | 1.52E−02 |
| 30 | KJX08128.1 \| | Periplasmic protein | +14.29 | −1.78 | 1.69E−05 |
| 16 | KPU05892.1 \| | Superoxide dismutase | +8.14 | −1.16 | 2.94E−02 |
| 32 | OUF11927.1 \| | Malate dehydrogenase | +7.45 | +1.43 | 1.17E−09 |
| 23 | AQT90079.1 \| | Outer membrane protein OmpX | +6.98 | −1.22 | 1.35E−07 |
| 41 | KTI01411.1 \| | Hypothetical protein ASV14_00320 | +6.63 | −1.17 | 2.53E−07 |
| 38 | ADF61881.1 \| | Superoxide dismutase | +5.94 | −1.05 | 1.01E−02 |
| 2 | KGB06270.1 \| | Catalase HPII | +5.02 | +2.33 | 2.89E−04 |
| 56 | KPU06091.1 | Molecular chaperone GroEL | +4.59 | +1.33 | 3.85E−06 |
| 1 | KGB06270.1 \| | Catalase HPII | +3.41 | +1.76 | 3.42E−03 |
| 15 | KGB03490.1 \| | DNA protection during starvation protein | +3.00 | −3.74 | 1.39E−02 |
| 14 | KLG03309.1 \| | Peroxiredoxin OsmC | +2.74 | −2.44 | 6.58E−03 |
| 59 | KPU06091.1 \| | Molecular chaperone GroEL | +2.21 | +1.36 | 1.56E−02 |
| 76 | SAC00082.1 \| | LysM domain/BON superfamily protein | +2.07 | −1.52 | 5.26E−06 |
| 65 | SAB00191.1 \| | Tellurium resistance protein TerD | +1.90 | −1.61 | 3.66E−05 |
| 78 | KPU04452.1 \| | Ecotin | +1.87 | −1.56 | 1.79E−08 |
| 77 | KJC00085.1 \| | Glyceraldehyde-3-phosphate dehydrogenase | +1.71 | −2.55 | 3.38E−07 |
| 70 | KGZ01983.1 \| | Chemical-damaging agent resistance protein C note like to terD domain protein and Tellurium resistance protein TerE | +1.62 | −1.50 | 1.71E−01 |
| 48 | KPU01695.1 \| | Elongation factor Tu | +1.47 | −1.55 | 2.09E−03 |
| 35 | KGB04009.1 \| | Bacterial extracellular solute-bindings, 3 family protein | +1.38 | −1.91 | 6.37E−04 |

DIGE (Difference in Gel Electrophoresis) was conducted in this experiment to overcome the gel-to-gel variation, which might be resulted from gel casting, running and staining. The system is based on the specific properties of CyDye$^{TM}$ DIGE Fluor dyes that enable multiplexing of separate protein mixtures on the same 2-D gel. The mixing sample with internal standard (pooled samples) technique allows for multivariable analyses, as samples can be separated and compared on several DIGE gels that were co-ordinated by the internal standard. The ability to directly compare two samples on the same gel not only avoids the complications of gel-to-gel variation but also enables a more accurate and rapid analysis of differences and reduces the number of gels that need to be run (*Westermeier & Scheibe, 2008*). Up- or down-regulation of protein expression among these three strains is summarized in Tables 3–5 and the presumtive conclusion can be emphasized as followings.

## Catalase isoforms serve as common adaptive responses of *E. cloacae* upon exposure to toxic cadmium

Cadmium ions exert their toxicity on the growth of *E. cloacae* in different degree of inhibition, depending on the protein expression in each strain (Fig. 1). These toxic effects possibly derived from the generation of reactive oxygen and nitrogen species (ROS

**Table 5  Regulation of major responsive proteins found in EC07 (cut-off at 1.3 fold).**

| Spot no. | Accession no. | Protein names | Ratio of 07-Cd/01-Cd | Ratio of 07-Cd/07-control | Anova |
|---|---|---|---|---|---|
| 25 | SAF00211.1 | | Protein YciF | 84.7 | −1.06 | 4.32E−09 |
| 45 | SAF58867.1 | | OmpC porin | 53.2 | 1.38 | 8.59E−07 |
| 71 | SAF00152.1 | | Manganese catalase | 51.7 | 1.13 | 3.21E−05 |
| 26 | OUF00075.1 | | Osmotically-inducible protein Y | 42.8 | 1.10 | 3.72E−03 |
| 75 | Q84FI1.1 |DPS_ENTCL | DNA protection during starvation protein | 32.7 | −1.21 | 8.29E−07 |
| 18 | SAH00153.1 | | OsmC family protein | 29.2 | −1.91 | 5.04E−08 |
| 8 | KJC00120.1 | | Hydroperoxidase II | 24.2 | −1.87 | 2.82E−02 |
| 27 | CZX11689.1 | | Protein YciF | 19.8 | 2.45 | 3.65E−05 |
| 4 | KGB06270.1 | | Catalase HPII | 18.5 | 1.21 | 1.02E−05 |
| 11 | KJC00120.1 | | Hydroperoxidase | 13.9 | 3.88 | 3.11E−04 |
| 19 | SAG00330.1 | | DNA protection during starvation protein | 12.3 | 2.01 | 4.55E−05 |
| 47 | SAH00347.1 | | Alcohol dehydrogenase | 11.7 | −1.15 | 2.21E−05 |
| 43 | KJX05875.1 | | Porin | 10.1 | 1.27 | 5.81E−09 |
| 20 | SAG00330.1 | | DNA protection during starvation protein | 9.9 | −5.1 | 4.68E−07 |
| 42 | CZW04458.1 | | Outer membrane protein A | 6.6 | 1.24 | 4.62E−08 |
| 51 | CZW60501.1 | | Isocitrate dehydrogenase | 5.4 | −1.51 | 1.49E−08 |
| 5 | KJC00120.1 | | Hydroperoxidase | 3.7 | −2.04 | 1.70E−07 |
| 54 | WP_075208257.1 | | Translation elongation factor Tu | 3.6 | −1.84 | 2.35E−05 |
| 9 | KJX08087.1 | | Succinate-semialdehyde dehydrogenase | 3.4 | −1.13 | 7.38E−10 |
| 60 | KJJ00126.1 | | Aspartate ammonia-lyase | 3.4 | −1.37 | 9.13E−06 |
| 55 | KJJ00122.1 | | Molecular chaperone GroEL | 3.1 | −1.29 | 7.48E−06 |
| 72 | KJJ00123.1 | | Molecular chaperone GroES | 2.8 | 1.27 | 1.02E−05 |
| 44 | 5FVN |A | Chain A, X-ray Crystal Structure Of *Enterobacter cloacae* Ompe36 Porin | 2.6 | 1.09 | 2.01E−07 |
| 24 | SAF00184.1 | | Protein YciE | 2.4 | −1.74 | 2.68E−04 |
| 29 | SAD00058.1 | | TrpR binding protein WrbA | 2.0 | 1.41 | 8.72E−10 |
| 46 | KJQ00573.1 | | Fructose-bisphosphate aldolase | 2.0 | −1.18 | 3.50E−03 |
| 10 | KTI00918.1 | | Dihydrolipoamide dehydrogenase | 2.0 | −1.05 | 1.66E−06 |
| 61 | SAB00249.1 | | 30S ribosomal protein S1 | 1.9 | 1.03 | 2.31E−03 |
| 73 | KJJ00123.1 | | Molecular chaperone GroES | 1.8 | 1.29 | 4.92E−03 |
| 31 | CZW02624.1 | | Periplasmic binding protein/LacI transcriptional regulator | 1.7 | −1.34 | 6.04E−09 |
| 67 | WP_004144995.1 | | MULTISPECIES: ATP synthase subunit alpha | 1.6 | −1.19 | 4.42E−03 |
| 62 | SAJ00217.1 | | F0F1 ATP synthase subunit beta | 1.6 | −1.14 | 2.46E−04 |
| 52 | SAH00247.1 | | Enolase | 1.5 | −1.35 | 1.65E−03 |
| 33 | CZX00043.1 | | 2,5-diketo-D-gluconate reductase A | 1.4 | −1.07 | 1.46E−02 |
| 49 | KPU01695.1 | | Elongation factor Tu | 1.4 | −1.46 | 4.33E−02 |
| 53 | SAI00778.1 | | Elongation factor Tu | 1.3 | −1.8 | 3.38E−02 |

and RNS) (*Brennan & Schiestl, 1996*; *Isarankura-Na-Ayudhya et al., 2018*; *Stohs & Bagchi, 1995*). Such free radical was not mainly produced by cadmium itself, however, indirect formation of ROS and RNS e.g., superoxide radical, hydroxyl radical and nitric oxide has been reported (*Waisberg et al., 2003*). Generation of non-radical hydrogen peroxide, which

| Condition | *DUF326* gene Mean | S.D. | *GAPDH* gene Mean | S.D. | ΔCq control | ΔCq cadmium | ΔΔCq | Fold difference |
|---|---|---|---|---|---|---|---|---|
| EC01 control | 23.28 | 4.24 | 31.61 | 4.30 | -8.33 | - | -3.82 | 14.13 |
| EC01 cadmium | 21.49 | 1.49 | 26.00 | 2.72 | - | -4.51 | | |
| EC07 control | 32.57 | 1.87 | 34.88 | 2.36 | -2.31 | - | -0.35 | 1.28 |
| EC07 cadmium | 31.16 | 1.53 | 33.11 | 1.71 | - | -1.96 | | |
| ATCC control | 31.56 | 0.99 | 25.24 | 2.86 | 6.32 | - | -0.57 | 1.49 |
| ATCC cadmium | 33.59 | 2.39 | 26.70 | 1.49 | - | 6.89 | | |

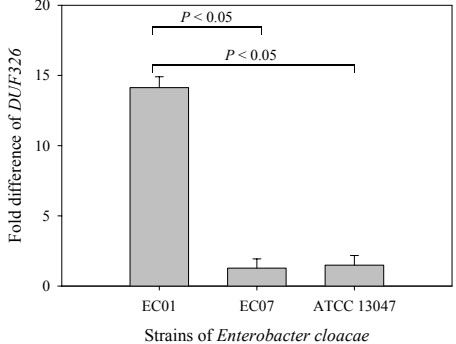

**Figure 7** **Fold differences in transcription level between *DUF326* gene (target gene) and *GAPDH* gene (housekeeping gene) in *E. cloacae* strains EC01, EC07 and ATCC 13047 in the absence and presence of cadmium ions.**

itself may in turn be a significant source of radicals *via* Fenton chemistry, was thought to be involved in this process. Furthermore, cadmium was also proposed to replace copper and iron in various cytoplasmic and membrane proteins such as ferritin, apoferritin. This resulted in the increase amount of free copper and iron, which consequently participated in oxidative stress *via* Fenton reactions (*Price & Joshi, 1983*). In this study, the catalase was found to be the only one protein shared among all three strains in the presence of cadmium stress (Fig. 6). Enhancement of catalase activity was also observed in the EC strain B1 when cultured in the cadmium condition (*Banerjee et al., 2015*). Importantly, the catalase has been proven to be the major enzyme used to detoxify these harmful effects in the case of ATCC 13047 (Fig. 8C). Even though the tolerant strain (EC01) did not use the catalase as the primary responses, it showed a specific pattern of catalase expression (spots no. 1, 2, 6 and 7). It should also be noted that spot no. 8 was found to be unique in the intolerant strain (EC07) (Table 3). Supportive evidences were documented on the presence of different isoforms of catalase found in *Pseudomonas aeruginosa* and *P. stutzeri* under various growth conditions such as under starvation conditions, induced by exposure to S-nitrosoglutathione (nitric oxide-generating reagent), and induced by exposure to sodium nitroprusside (*Kawakami et al., 2010*; *Xie et al., 2014*). In addition, *Xanthomonas campestris* responded against cadmium stress by regulation of two monofunctional catalase isozymes (*Banjerdkij, Vattanaviboon & Mongkolsuk, 2005*). Therefore, up-regulation of

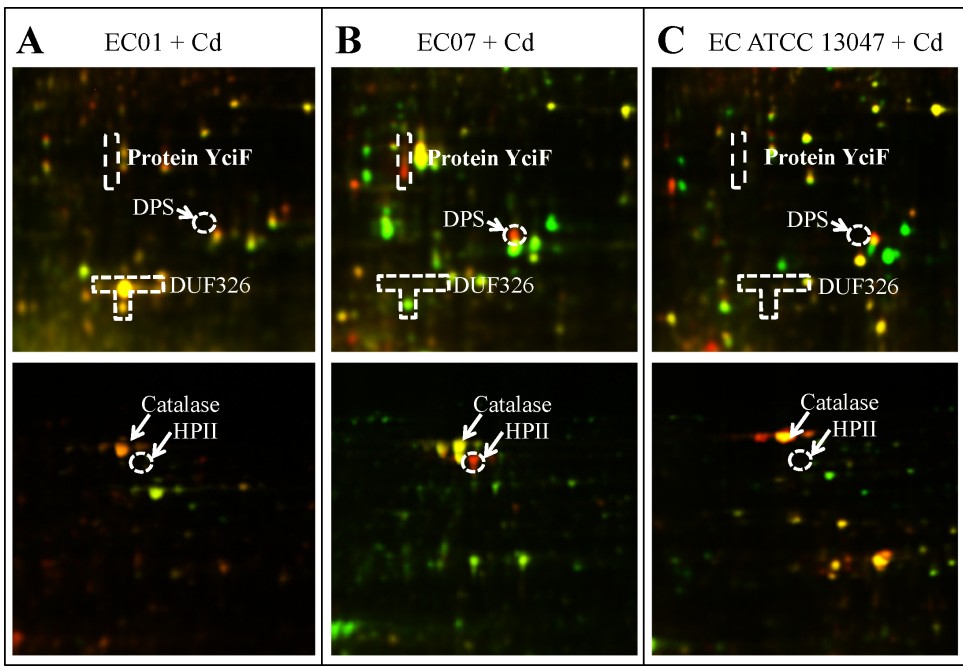

**Figure 8** The superimposed of 2D-DIGE gels representing differentially expressed proteins of *E. cloacae* strains EC01 (A), EC07 (B) and ATCC 13047 (C) in the presence of cadmium ions at different locations (zoomed gels).

catalase found herein was proposed not only to associate with an adaptation of *E. cloacae* in survival under oxidative stress but also to reduce ROS in the cells due to Cd actions. Moreover, the bacteria selected different isoforms of catalase in manner of Cd responses.

## DUF326-like domain: a novel molecule participated in detoxification of cadmium stress

It has been suggested that the EC01 could tolerate to toxic cadmium by expressing cadmium-binding proteins (Fig. 3). This coincided with the high level expression of the DUF326-like domain (Table 4 and Fig. 8A). To our knowledge, this is the first report on the involvement of DUF326-like domain in detoxification of cadmium stress in bacteria since the world-wide availability of molecular function of DUF326 is quite limited (http://www.ebi.ac.uk/interpro/entry/IPR005560). When the DUF326-like domain was subjected to blast on https://blast.ncbi.nlm.nih.gov/Blast.cgi, the results showed 100% identity with four helix bundles copper-binding protein (Fig. 9) of Family 'Enterobecteriaceae'. The copper-binding protein is a cysteine-rich protein that can be found in many copper resistant bacteria. Previous study demonstrated that *Pseudomonas putida* expressed three Cd-binding proteins (CdBP1, CdBP2 and CdBP3), which contain large amounts of cysteine residue. Among these, the CdBP3 has the highest cysteine content of 22% (*Higham, Sadler & Scawen, 1983*). Compared to our finding, the DUF326 protein comprises of 19 cysteine residues from 108 amino acid (17.6%) (Fig. 9). It can be suggested that the thiol groups derived from these cysteine-rich proteins provide

MLDEFKKCIESCYLCAVACDHCAASCLEEENLEMMRECIKLDMQCANICRLAAQ
FMALNSESARELCRVCADVCQQCGDECAKHEHEHCQNCSRACHHCAEQCRRMAA

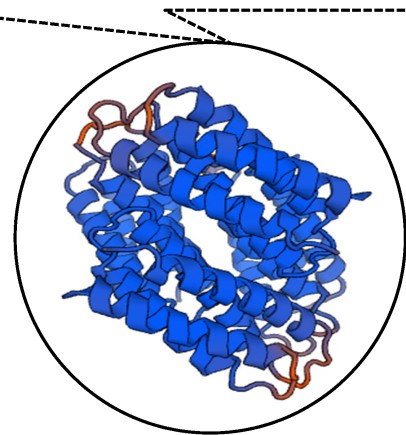

**Figure 9** **Amino acid sequences and the predicted tertiary structure of a domain of uncharacterized function (DUF326).**

very high binding avidity to Cd. Additionally, protein sequence analysis by searching on UniProtKB database described that the cysteines mostly follow a C-X(2)-C-X(3)-C-X(2)-C-X(3) pattern, though they often appear at other positions in the repeat as well (https://www.ebi.ac.uk/interpro/entry/IPR005560). The result from domain architecture mode analysis had shown two domain types including 4Fe-4S ferredoxin-type, iron-sulphur binding domain (IPR017896) found in the DUF326. Similar observation was also found in the DUF59 that involved in FeS protein maturation and/or intracellular Fe homeostasis (*Mashruwala & Boyd, 2018*). Supportive evidences on the interconnection between DUF326 and 4Fe-4S ferredoxin protein (YwjF) and other related iron-sulfur containing proteins were identified by the STRING database as shown in Fig. 10. Overexpression of DUF326-like domain was believed to protect the *E. cloacae* from hazardous cadmium ions, possibly by high binding affinity to cadmium. It has extensively been reported the imperative roles of domain of unknown function on stress responses in many organisms as follows. In *P. aeruginosa*, induction of PA1994 (classified as a member of DUF1089) was proposed to be responsive for the bacterial cell-wall stress or host-pathogen interactions (*Bakolitsa et al., 2010*). The DUF1471 found in many bacteria in the Enterobacteriaceae played roles in stress response, biofilm formation, and pathogenesis (*Eletsky et al., 2014*). In fungus, the DUF1996 regions involved in the regulation of multiple stress responses and environmental adaptation (*Tong et al., 2016*). In rice, the DUF1644 and DUF966 genes were regulated in response to drought and salinity stresses (*Guo et al., 2016*; *Luo et al., 2014*).

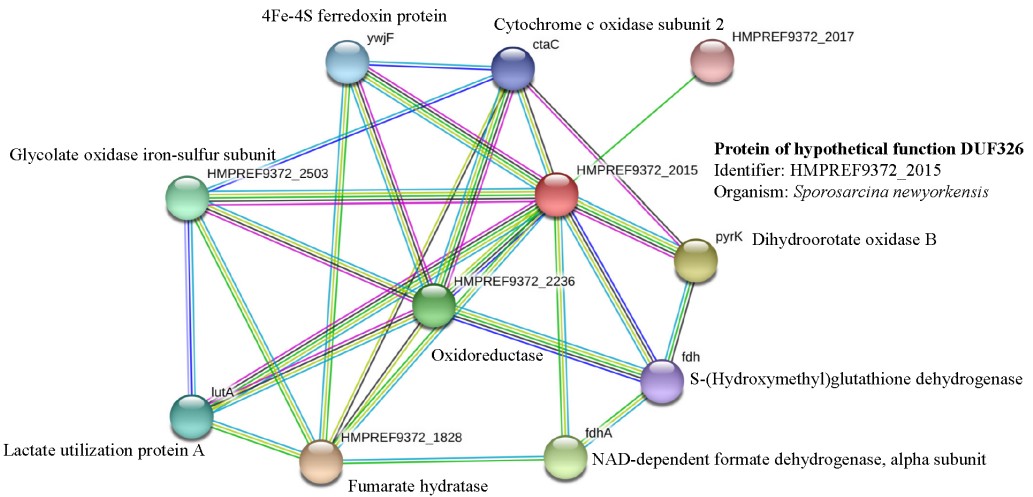

**Figure 10** Relationship between genes encoding protein of hypothetical function DUF326 (from *Sporosarcina newyorkensis*) and other proteins as identified by the STRING software.

## Protein networks involved in the survival of intolerant strain of *E. cloacae* in cadmium stress

In case of EC07, the top-ranked of protein expression in survival against cadmium stress included the YciF protein, OmpC porin, manganese catalase, Osmotically-inducible protein Y, DNA protection during starvation protein, and OsmC family protein (Table 5). Expression of the YciF protein, which is a stress-inducible protein, was found to be in the highest level. The structure and function of this protein have not extensively been studied. After searching in UniProtKB/TrEMBL database, the YciF might contain some components closed to the DUF892 family that possessed iron sequestration funtion (*Liu et al., 2016*). It is possible that the YciF is up-regulated when the bacteria encounter stress conditions. From the crystal structure of YciF protein of *Escherichia coli* (*Hindupur et al., 2006*), a hypothesis can be emphasized that the YciF may possess metal-binding properties since its structure is similar to others metal-binding proteins such as rubrerythrin (a protein that has a di-iron center), ferritin and monooxygenases (*Hindupur et al., 2006*). The metal-binding sites are proposed to lie in the helix bundle within the two pockets (P1 and P2). The architectural features of YciF are also found to associate with other protein functions such as the manganese catalase domain in the protein bll3758 from *Bradyrhizobium japonicum* (*Hindupur et al., 2006*).

Using the STRING software, the YciF protein has a direct interconnection with the OsmC (encoding peroxiredoxin) (Fig. 11). In addition, it has some indirect connection with a series of outer membrane proteins (OmpC, OmpA and OmpX) linked by the YdeI protein (induced by cadmium, hydrogen peroxide and acid stress) (*Lee et al., 2010*). The OmpC porin is known to involve in biological process as ion transport system. This protein has been reported mostly to participate in antibiotic resistant mechanism of *E. cloacae*. Disruption of OmpC rendered the cells to be resistant against carbapenem (*Uechi et al., 2019*). However, little is known about the correlation between cadmium stress and alteration

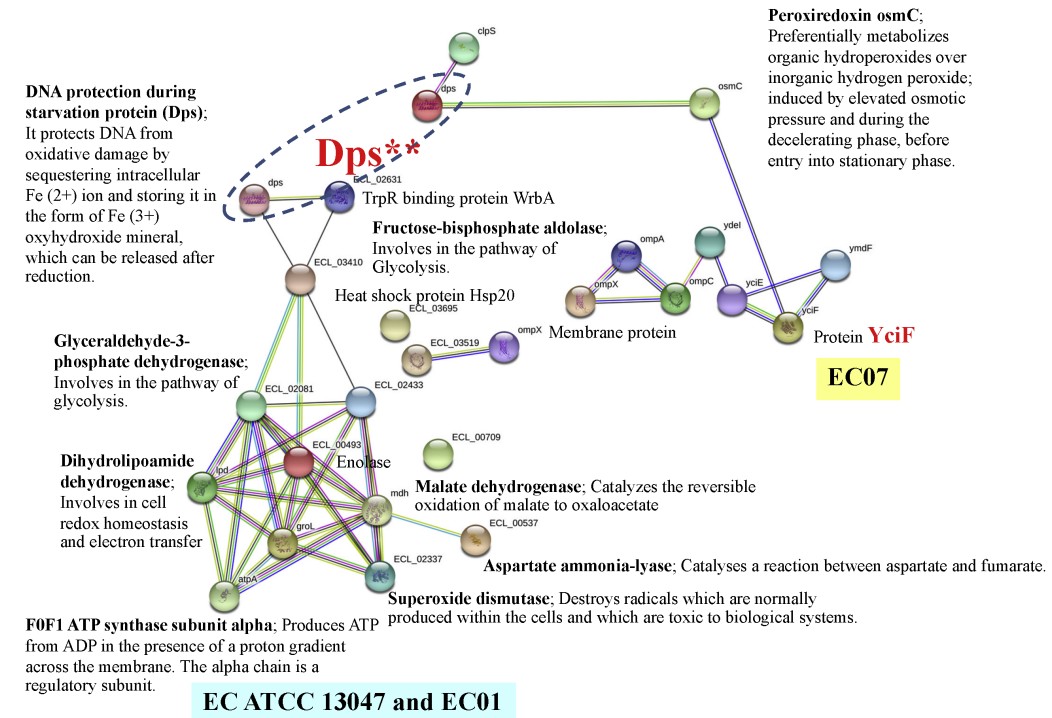

**Figure 11** Relationship between genes encoding proteins of *E. cloacae* strains EC01, EC07 and ATCC 13047 involved in responses to cadmium stress as identified by the STRING software.

in OmpC porin. In the present study, the fold difference in expression of OmpC porin between the EC07 and EC01 was approximately 53 fold (Table 5). Similar observation was also found in *E. coli* in which deletion of OmpC gene led to decreased cadmium resistance (*Egler et al., 2005*). Additionally, it is noteworthy that there is a linkage between the OsmC and Dps proteins. More importantly, there is a high possibility that the Dps might be a connecting junction with the other proteins involved in the metabolic processes (glycolysis and tricarboxylic acid cycle), ATP production, chaperone and oxidative scavenging enzyme.

The increase of Dps under cadmium-stress condition might be associated with DNA protection of *E. cloacae* to survive. A previous study mentioned that the levels of cellular DNA damage were significantly higher in cadmium adding in cultures than in controls (*Shapiro & Keasling, 1996*). Explanation can be drawn that a mutagenic effect by cadmium is not due to direct interaction with DNA but rather to inhibit DNA repair systems (*Serero et al., 2008*). The Dps was also reported to be expressed during stationary phase upon starvation or exposure to severe environmental assaults, including oxidative stress and nutritional deprivation (*Calhoun & Kwon, 2011*; *Nair & Finkel, 2004*). It could effectively bind with chromosomal DNA and form the stable Dps-DNA co-crystal. The mutants of Dps also failed to develop starvation-induced resistance to hydrogen peroxide, an agent that can cause oxidative damage to DNA, and show dramatic changes in the pattern of proteins synthesis during starvation in *E. coli* (*Almiron et al., 1992*). Such protection can be mediated by the intrinsic properties such as DNA binding, iron sequestration and ferroxidase activity

(*Calhoun & Kwon, 2011*). These properties lend support to the importance of Dps in iron- and hydrogen peroxide-detoxification and acid resistance. Moreover, the prevalence of Dps and Dps-like proteins in bacteria suggests that protection involving DNA and iron sequestration is crucial and widespread in prokaryotes (*Calhoun & Kwon, 2011*). The Dps has also been identified as a ferritin-like protein, partly because of its ferroxidase activity, or more specifically, its ability to oxidize bound ferrous ions to the ferric state and storing it in the form of $Fe^{3+}$ ferrihydrite mineral, which can be released after reduction (*Calhoun & Kwon, 2011*; *Nair & Finkel, 2004*; *Velayudhan et al., 2007*). In fact, the primary role of Dps may not be iron storage, but to protect macromolecules from the combined lethality of ferrous ions and $H_2O_2$ (*Abdul-Tehrani et al., 1999*) and by the fact that Dps strongly prefers $H_2O_2$ over $O_2$ as an oxidant when preparing iron for storage. The catalase activity of Dps seems to be significant in Dps-mediated protection against hydrogen peroxide stress, as *Zhao et al. (2002)* has identified its ability to decompose hydrogen peroxide as a significant method of detoxification in *E. coli*. Altogether, the protective roles of Dps are most likely achieved through various functions associated with the protein-DNA binding and chromosome compaction, metal chelation, ferroxidase activity, regulation of gene expression and a weak catalase activity.

Glyceraldehyde-3-phosphate dehydrogenase (GAPDH) is an enzyme of ~37 kDa that catalyzes the sixth step of glycolysis and thus serves to break down glucose for energy and GAPDH is also involved carbon molecules. In addition to this long established metabolic function, GAPDH has also involved in activation and initiation of apoptosis (*Tarze et al., 2007*). Metabolic adaptation and energy production are crucial for survival under heavy metal load/stress. This enzyme facilitates the enhanced metabolic needs for survival followed by an enhanced expression of proteins (*Vranakis et al., 2014*). Additionally, the alterations in the flux of metabolites can create metabolic networks, allowing an organism to go along in a changing environment (*Mailloux, Lemire & Appanna, 2011*). The GAPDH and related enzymes were also shown to be involved in the adaptive response to oxidative stress (*Oh et al., 2002*; *Ralser et al., 2007*). It can also act as a reversible metabolic switch under oxidative stress. The temporary inactivation of GAPDH by oxidant treatment may re-route the metabolic flux from glycolysis to the pentose-phosphate pathway, allowing the cell to generate more NADPH as an antioxidant cofactor (*Ralser et al., 2007*).

## CONCLUSIONS

With respect to the performances of proteomics and bioinformatics tools, this work successfully explores that the DUF326-like domain of *Enterobacter cloacae* strain EC01 (cadmium-tolerant strain) has been found to play imperative roles in detoxification of cadmium stress, possibly by binding to cadmium ions. Such protective effect helps to recover the rate of cell division particularly at the middle- to late-log phase of growth curve in the presence of cadmium ions. This also coincides with the capability to accumulate cadmium ions intracellularly at 6 h. The remarkable binding capability due to the presence of cysteine-rich domains of DUF326-like domain has been observed in the case of EC01 of approximately 2 fold higher than those of the EC07 (cadmium-intolerant strain) and ATCC 13047. Expression of the DUF326-like domain is more pronounced of up to 220-fold

higher than the others. Results from the qRT-PCR confirm that the transcription level of *DUF326* gene in the EC01 is approximately 14-fold higher than those of the others. This indicates that the cadmium-tolerant strain of EC adapts itself by complexation of cadmium ions by metal-binding proteins rather than using the proteins/enzymes involved in oxidative deterioration, stress responses or outer membrane portion as observed in the case of intolerant or standard strains. Taken altogether, our findings explore the molecular mechanisms of cadmium tolerance in EC and also open up a high possibility of applying either the DUF326-like domain or the EC01 cells as potential tools for cadmium bioremediation in the future.

### Funding

This research is financially supported in part by the Young Research Scholar grant (no. MRG5480205) from The Thailand Research Fund, the grant from National Research Council of Thailand (B.E. 2561-2562) to Patcharee Isarankura-Na-Ayudhya, and the annual governmental grant of Mahidol University (B.E. 2557-2559) to Chartchalerm Isarankura-Na-Ayudhya. There was no additional external funding received for this study. The funders had no role in study design, data collection and analysis, decision to publish, or preparation of the manuscript.

### Grant Disclosures

The following grant information was disclosed by the authors:
Young Research Scholar: MRG5480205.
National Research Council of Thailand: B.E. 2561-2562.
Mahidol University: B.E. 2557-2559.

### Competing Interests

The authors declare there are no competing interests.

### Author Contributions

- Kitipong Chuanboon performed the experiments, analyzed the data, prepared figures and/or tables, authored or reviewed drafts of the paper.
- Piyada Na Nakorn, Supitcha Pannengpetch and Vishuda Laengsri performed the experiments.
- Pornlada Nuchnoi conceived and designed the experiments, analyzed the data, contributed reagents/materials/analysis tools.
- Chartchalerm Isarankura-Na-Ayudhya conceived and designed the experiments, analyzed the data, contributed reagents/materials/analysis tools, prepared figures and/or tables, authored or reviewed drafts of the paper.
- Patcharee Isarankura-Na-Ayudhya conceived and designed the experiments, performed the experiments, analyzed the data, contributed reagents/materials/analysis tools, prepared figures and/or tables, authored or reviewed drafts of the paper, approved the final draft.

## Data Availability

Data are deposited in the repository room at the Faculty of Medical Technology, Mahidol University, Salaya campus, Nakornpathom, Thailand. This repository has been recognized by the Center for Occupational Safety, Health and Environment Management (COSHEM), Mahidol University and the Bureau of Laboratory Quality Standards, Department of Medical Sciences, Ministry of Public Health, Thailand.

Deposition reference numbers: EC_B-Cd-A series. Strains are also available upon request from the corresponding author.

## Supplemental Information

Supplemental information for this article can be found online at http://dx.doi.org/10.7717/peerj.6904#supplemental-information.

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
