# Peer review of "Proteomics and bioinformatics analysis reveal potential roles of cadmium-binding proteins in cadmium tolerance and accumulation of Enterobacter cloacae"

_PeerJ, doi:10.7717/peerj.6904_

## Round 0.1 · original submission · Minor Revisions

Please address all critiques of both reviewers and revise your manuscript accordingly.

Reviewer 1 ·

Basic reporting

In this manuscript, the authors report the potential cadmium-binding proteins in Enterobacter cloacae, a gram-negative bacteria. The manuscript is well-written and the authors provide sufficient background information and discussed the results clearly for the understanding to broad interest readers.

Experimental design

The authors provide a complete detail description of the methods in the manuscript.

Validity of the findings

No comment

Additional comments

Few Minor issues:
1. The title of the manuscript is too long. It should have a clear, precise scientific meaning.
2. Change "folds" to "fold" in the manuscript.
3. I suggest improving the introduction section; should be more clear. Some sentences in the introduction section seem to be incomplete.
3. Re-write the line 168-171: not clear.
4. Explain in detail the method used for the estimation of cadmium accumulation in the bacterial as mentioned in line 183-184.
5. Line204: Change ' labeling' to 'labeled'.
6. In Figure 4 legend, the % of protein gel written is 12% whereas in the text it is mentioned as 12.5%. Write the correct % of gel used.
7. The authors mentioned all the 80 proteins spots in Table 3. I suggest to include a graph also showing the number of the protein identified on y-axis versus protein family on x-axis for easy understanding of the result.
8. Similarly, include a graph showing the max fold change between control and cadmium-treated group for qRT-PCR result as shown in Table 6.
9. Re-write the line 486-488; not clear.

Reviewer 2 ·

Basic reporting

The authors Chuanboob et al. have presented a detailed study of cadmium-binding proteins involved in inducing cadmium tolerance in different species of Enterobacter cloacae isolated from industrial and agricultural sites around central Thailand. The study is well-designed, and the authors have presented conclusive results corroborating their claims. The manuscript is easy to follow with overall correct use of language and grammar. The manuscript is structured as per PeerJ requirements and the authors have presented relevant and adequate background information and references to set up their research question.

Experimental design

The figures and tables presented corroborate the conclusion made by the authors. The reviewer is thankful to the authors for making the necessary raw data available. The authors have following few suggestions that will improve the quality of manuscript
1. The figures are of high quality, easy to follow, and have the necessary legends. The authors should please consider adding legend to figure 1. Though the figure description is adequate, having a legend indicating what open and closed symbols mean within the image will make it very easy for readers to follow.
2. The Venn diagram presented in figure 5 is insightful in differentiating protein expression in the three E.c species. As per the diagram, there are no common protein(s) between ATCC 13047 and intolerant strains used in the study. It is a bit surprising as there is 1 common protein between the tolerant and intolerant strains. Can the authors elaborate or provide an explanation for the same?

Validity of the findings

The authors have presented a solid data supporting their claims about the mechanism of cadmium tolerance observed in EC01 strain due to over expression of cysteine-rich DUF326. The study is suited for publication in PeerJ with a few minor suggestions mentioned above.

---

## Round 0.2 · accepted · Accept

All critical points were adequately addressed and the manuscript was amended accordingly. Therefore, revised version is acceptable in its current form.


#